# How to deal with missing data in supervised deep learning?

**Niels Bruun Ipsen**[*]
nbip@dtu.dk

**Pierre-Alexandre Mattei**[†‡]
pierre-alexandre.mattei@inria.fr

**Jes Frellsen**[*‡]
jefr@dtu.dk

## Abstract

The issue of missing data in supervised learning has been largely overlooked, especially in the deep learning community. We investigate strategies to adapt neural architectures for handling missing values. Here, we focus on regression and classification problems where the features are assumed to be missing at random. Of particular interest are schemes that allow reusing as-is a neural discriminative architecture. To address supervised deep learning with missing values, we propose to marginalize over missing values in a joint model of covariates and outcomes. Thereby, we leverage both the flexibility of deep generative models to describe the distribution of the covariates and the power of purely discriminative models to make predictions. More precisely, a deep latent variable model can be learned jointly with the discriminative model, using importance-weighted variational inference, essentially using importance sampling to mimick averaging over multiple imputations. In low-capacity regimes, or when the discriminative model has a strong inductive bias, we find that our hybrid generative/discriminative approach generally outperforms single imputations methods.

## 1 Introduction

Missing data affect data analysis across a wide range of domains and the sources of missing values span an equally wide range. Recently, deep latent variable models (DLVMs, Kingma & Welling, 2014; Rezende et al., 2014) have been applied to missing data problems in an unsupervised setting (e.g. Rezende et al., 2014; Nazabal et al., 2020; Ma et al., 2018; 2019; Ivanov et al., 2019; Mattei & Frellsen, 2018; 2019; Yoon et al., 2018; Li et al., 2019; Ipsen et al., 2021; Ghalebikesabi et al., 2021), while the supervised setting has not seen the same attention. The progress in the unsupervised setting is focused on inference and imputation and can therefore be useful as an imputation step before learning a discriminative model. Traditionally the focus has also been on inference and imputation either as a goal in itself or before supervised learning on the (possibly multiple) imputations and observed data (Rubin, 1976; 1996; Little & Rubin, 2019).

Supervised learning in the presence of missing values has different goals and pose different challenges than inference and imputation (Josse

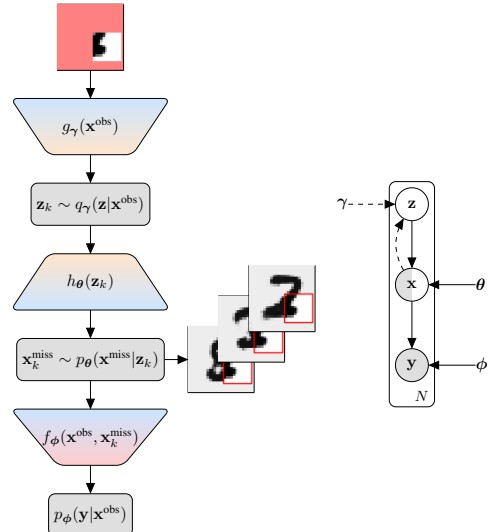

Figure 1: **Left:** supMIWAE computational structure: encoding network $g_{\gamma}$, decoding network $h_{\theta}$ and discriminative network $f_{\phi}$. **Right:** Graphical model: supMIWAE seen as a VAE for missing values concatenated with neural discriminator.

[*]Department of Applied Mathematics and Computer Science, Technical University of Denmark, Denmark
[†]Université Côte d'Azur, Inria (Maasai team), Laboratoire J.A. Dieudonné, CNRS, France
[‡]Equal contribution

et al., 2019; Le Morvan et al., 2020a; 2021). Here the overall aim is to minimize an expected prediction error. However, optimal single imputation does not necessarily lead to optimal prediction in terms of minimizing a prediction error (Josse et al., 2019). One challenge is that predictions based on inputs with missing values can be ambiguous, that is, the conditional distribution of the missing values given the observed may be multimodal and the optimal prediction may change with the mode. With single imputation, the conditional distribution over the missing data is discarded, and the optimal single imputation can no longer reflect this ambiguity, see figure 2. Even the optimal single imputation leads to biased parameter estimates compared to the fully observed case (Bertsimas et al., 2021), and may in some cases lie outside the distribution of the data. Another challenge is that the number of missing value patterns grows combinatorially with the number of features $p$, requiring $2^p$ submodels to fit the Bayes predictor in the linear case (Le Morvan et al., 2020a). In multiple imputation (Rubin, 1996), several imputations are drawn from the posterior predictive distribution of the missing values, reflecting the full conditional distribution of the missing values and thus the uncertainty about what is missing. This allows for uncertainty estimates in downstream tasks such as prediction. This in turn requires fitting as many discriminative models as the number of imputations.

## 1.1 CONTRIBUTIONS

In order to address supervised deep learning with missing values we develop the *supervised missing data importance-weighted autoencoder (supMIWAE) bound*, based on the approximate maximum likehood techniques used by Burda et al. (2016) and Mattei & Frellsen (2019). This is a scalable approach to marginalizing over missing values in a joint model of covariates and outcomes

$$p_{\phi,\theta}(\mathbf{y}, \mathbf{x}^{\text{obs}}, \mathbf{x}^{\text{miss}}) = p_\phi(\mathbf{y}|\mathbf{x}^{\text{obs}}, \mathbf{x}^{\text{miss}})p_\theta(\mathbf{x}^{\text{obs}}, \mathbf{x}^{\text{miss}}). \tag{1}$$

The covariate model $p_\theta(\mathbf{x}^{\text{obs}}, \mathbf{x}^{\text{miss}}) = \int p(\mathbf{x}^{\text{obs}}, \mathbf{x}^{\text{miss}}|\mathbf{z})p(\mathbf{z})\, d\mathbf{z}$ is a DLVM in this work, but can be any probabilistic model imposing a joint density over covariates. The outcome model $p_\phi(\mathbf{y}|\mathbf{x}^{\text{obs}}, \mathbf{x}^{\text{miss}})$ is any neural discriminative architecture that would have been used in the complete-data case, parameterizing a density over the outcomes. The graphical model is shown in figure 1 along with its computational structure.

Once the joint model has been trained using the supMIWAE bound, the conditional distribution of the outcome given the observed parts of the input can be found as

$$p_\phi(\mathbf{y}|\mathbf{x}^{\text{obs}}) = \int p_\phi(\mathbf{y}|\mathbf{x}^{\text{obs}}, \mathbf{x}^{\text{miss}})p_\theta(\mathbf{x}^{\text{miss}}|\mathbf{x}^{\text{obs}})\, d\mathbf{x}^{\text{miss}}. \tag{2}$$

This is approximated using importance sampling techniques, averaging predictions over multiple importance samples (akin to multiple imputations, see appendix C for a deeper discussion) from the generative model.

The model can be trained end-to-end or a pretrained generative model can be coupled with a discriminative model. Having a joint model allows for supervised imputations, where available labels can help guide the imputations from the generative model by adjusting the importance weights accordingly.

Joint models over covariates and labels have previously been used to marginalize over missing values in the covariates, using Gaussian mixture models either as the model over the covariates or directly as the joint model over covariates and outcomes (Ghahramani & Jordan, 1995; Ghahramani & Hinton, 1996; Ahmad & Tresp, 1993; Tresp et al., 1994; 1995; Śmieja et al., 2018). Our approach shows how to use more flexible generative models, such as DLVMs, provides an efficient optimization procedure and allows for keeping any deep discriminative architecture unchanged.

## 2 BACKGROUND AND NOTATION

We define the random variable $\mathbf{x} = (x_1, \ldots, x_p) \in \mathcal{X}$ which takes values in a $p$-dimensional feature space $\mathcal{X} = \mathcal{X}_1 \times \cdots \times \mathcal{X}_p$. There is a corresponding (possibly vector valued) response variable $\mathbf{y} \in \mathcal{Y}$. A missing process obscures parts of $\mathbf{x}$ resulting in the mask random variable $\mathbf{s} \in \{0, 1\}^p$ where

$$s_j = \begin{cases} 1 & \text{if } x_j \text{ observed,} \\ 0 & \text{if } x_j \text{ missing.} \end{cases} \tag{3}$$

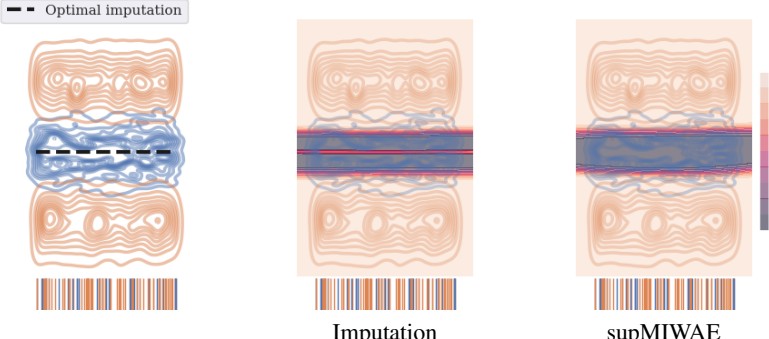

Figure 2: **Left:** Kernel density of a two-class dataset (the burger dataset), where and $\mathbf{y} = 0$ is blue and $\mathbf{y} = 1$ is brown/orange. Observations with missing values are shown as sticks in the bottom margin of the plots and colored according to their class label. For observations with missing 2nd coordinate there is an inherent ambiguity about the class with $p(\mathbf{y} = 1|\mathbf{x}^{\text{obs}}) = 2/3$. The optimal single imputation is shown as the dashed black line. **Middle:** Decision surface for a neural classifier trained on optimally-imputed data. The decision surface has to reflect that for the imputed data $p(\mathbf{y} = 1|\iota_0(\mathbf{x}^{\text{obs}})) = 2/3$, thus the increased class $\mathbf{y} = 1$ probability around the optimal imputations (here $\iota_0$ is the imputation function, see appendix A.1). **Right:** Decision surface as learnt by the supMIWAE. It is similar to the decision surface that would be found in the absence of missing data.

We let $\text{obs}(\mathbf{s})$ denote the indices of the non-zero entries of $\mathbf{s}$ and $\text{miss}(\boldsymbol{s})$ denote the indices of the zero-entries of $\mathbf{s}$, such that $\mathbf{x}^{\text{obs}(\mathbf{s})}$ is the set of observed elements of $\mathbf{x}$, and $\mathbf{x}^{\text{miss}(\mathbf{s})}$ is the set of missing elements of $\mathbf{x}$. For simplicity we will omit the $\mathbf{s}$ and write $\mathbf{x}^{\text{obs}}$, $\mathbf{x}^{\text{miss}}$ respectively, whenever the context is clear.

Following (Yoon et al., 2018; Le Morvan et al., 2020b; Ghalebikesabi et al., 2021) we define the incomplete random variable $\tilde{\mathbf{x}} = (\tilde{x}_1, \ldots, \tilde{x}_p) \in \tilde{\mathcal{X}}$ which takes values in $\tilde{\mathcal{X}} = (\mathcal{X}_1 \cup \{\texttt{na}\}) \times \cdots \times (\mathcal{X}_p \cup \{\texttt{na}\})$ where missing values are represented by the symbol $\texttt{na}$, such that $\texttt{na} \cdot x_j = \texttt{na}$ and $\texttt{na} \cdot 0 = 0$. We typically only have access to $\tilde{\mathbf{x}}$, where

$$\tilde{\mathbf{x}} = \mathbf{x} \odot \mathbf{s} + \texttt{na} \odot (1 - \mathbf{s}) \tag{4}$$

and $\odot$ is the Hadamard product. Finally, we are given $n$ i.i.d. copies of the random variables $\tilde{\mathbf{x}}$ and $\mathbf{y}$ which we collect in a dataset $\mathcal{D} = \{\tilde{\mathbf{x}}_i, \mathbf{y}_i\}_{i=1}^n$, or alternatively $\mathcal{D} = \{\mathbf{x}_i^{\text{obs}}, \mathbf{y}_i\}_{i=1}^n$.

We make the additional assumption that the data are *missing at random* (MAR, see e.g. Seaman et al., 2013; Little & Rubin, 2019). Specifically, this means that we assume that $\mathbf{s}$ and $\mathbf{x}^{\text{miss}}$ are independent given $\mathbf{x}^{\text{obs}}$. This assumption allows to avoid to explicitly model the missing mechanism. Our approach could be extended beyond MAR by using a generative model fit for this purpose, like the not-MIWAE of Ipsen et al. (2021), or the deep pattern-set mixture of Ghalebikesabi et al. (2021).

## 2.1 CHALLENGES WHEN TRAINING DISRIMINATIVE MODELS WITH MISSING DATA

Our predictive model will be defined through a mapping $f_\phi : \mathcal{X} \to H$ used to parameterize a conditional distribution

$$p_\phi(\mathbf{y}|\mathbf{x}) = \Psi(\mathbf{y}|f_\phi(\mathbf{x})). \tag{5}$$

Here $(\Psi(\cdot|\eta))_{\eta \in H}$ is a parametric family of distributions over the outcome space $\mathcal{Y}$, such as a Gaussian distribution in regression or a categorical distribution in classification. Under the maximum-likelihood framework (or equivalently the logarithmic scoring rule), an optimal mapping $f_\phi^*$ within a class $\mathcal{F} = (f_\phi)_{\phi \in \Phi}$ parameterized by $\phi \in \Phi$ is defined as

$$f_\phi^* \in \underset{f_\phi \in \mathcal{F}}{\arg\max} \, \mathbb{E}_{p^*(\mathbf{x}, \mathbf{y})} \left[ \log \Psi(\mathbf{y}|f_\phi(\mathbf{x})) \right], \tag{6}$$

where $p^*(\mathbf{x}, \mathbf{y})$ is the true data generating distribution. When the data are complete, this is typically approximated by maximizing the log likelihood of the parameters $\phi$ given the data $\mathcal{D}_{\text{train}}$,

$$\ell(\phi) = \sum_{(\mathbf{x}, \mathbf{y}) \in \mathcal{D}_{\text{train}}} \log p_\phi(\mathbf{y}|\mathbf{x}). \tag{7}$$

When the covariates have missing values the log-likelihood cannot be maximized directly as the likelihood is not defined since $p_\phi(\mathbf{y}|\mathbf{x})$ depends on the full input vector. The simplest approach is instead to learn separate mappings for each missing pattern. This reduces the amount of data available for the training of each mapping and does not scale well with the dimensionality of the input as in general $2^p$ networks need to be trained. An often used approach is instead to impute the missing values using a single imputation (for instance, using $\mathbb{E}_{p_\theta(\mathbf{x}^{\text{miss}}|\mathbf{x}^{\text{obs}})}[\mathbf{x}^{\text{miss}}]$ or an approximation of it), and then map the concatenation of observed and imputed values to approximate the conditional distribution

$$p_\phi(\mathbf{y}|\mathbf{x}^{\text{obs}}) \approx p_\phi(\mathbf{y}|\mathbf{x} = (\mathbf{x}^{\text{obs}}, \mathbb{E}_{p_\theta(\mathbf{x}^{\text{miss}}|\mathbf{x}^{\text{obs}})}[\mathbf{x}^{\text{miss}}])) = \Psi(\mathbf{y}|f_\phi(\mathbf{x}^{\text{obs}}, \mathbb{E}_{p_\theta(\mathbf{x}^{\text{miss}}|\mathbf{x}^{\text{obs}})}[\mathbf{x}^{\text{miss}}])). \quad (8)$$

Instead of $\mathbb{E}_{p_\theta(\mathbf{x}^{\text{miss}}|\mathbf{x}^{\text{obs}})}[\mathbf{x}^{\text{miss}}]$, which is the optimal imputation under the mean squared error, one could use any kind of imputation, for instance using a constant, or the mean.

This general approach is adequately called *impute-then-regress* by Bertsimas et al. (2021) and Le Morvan et al. (2021). While it will be Bayes-consistent given a powerful enough classifier (Le Morvan et al., 2021), it leads to biased parameter estimates compared to having complete data (Bertsimas et al., 2021). This is illustrated in figure 2 where the optimal (under the mean squared error) single imputations are placed entirely within one of the classes, requiring a biased decision surface to properly reflect the class label probabilities for the records with missing values. From a probabilistic perspective, computing $p_\phi(\mathbf{y}|\mathbf{x}^{\text{obs}})$ requires marginalizing over the missing features as in equation (1), that is

$$p_\phi(\mathbf{y}|\mathbf{x}^{\text{obs}}) = \mathbb{E}_{p_\theta(\mathbf{x}^{\text{miss}}|\mathbf{x}^{\text{obs}})}[p_\phi(\mathbf{y}|\mathbf{x}^{\text{obs}}, \mathbf{x}^{\text{miss}})]. \quad (9)$$

Notice the difference to single imputation since, in general

$$\mathbb{E}_{p_\theta(\mathbf{x}^{\text{miss}}|\mathbf{x}^{\text{obs}})}[p_\phi(\mathbf{y}|\mathbf{x}^{\text{obs}}, \mathbf{x}^{\text{miss}})] \neq p_\phi(\mathbf{y}|\mathbf{x} = (\mathbf{x}^{\text{obs}}, \mathbb{E}_{p_\theta(\mathbf{x}^{\text{miss}}|\mathbf{x}^{\text{obs}})}[\mathbf{x}^{\text{miss}}])), \quad (10)$$

except in some pathological cases (e.g. when $p_\phi$ is linear or $p_\theta$ is a Dirac distribution). Therefore, if the discriminative model is nonlinear and the generative model is complex enough, then using single imputation will give a very different result than marginalising over the missing features. This is exemplified in figure 2, where a classifier trained using the optimal imputation finds a solution that is very different to what would be found by one trained without missing data. Our technique, marginalizing the missing values, allows to recover the same decision surface as a classifier trained without missing data.

## 2.2 Our joint model and its supMIWAE variational bound

As mentioned in the introduction, we posit a joint model over covariates and outcome, that will make use of a latent variable $\mathbf{z}$:

$$p_{\phi,\theta}(\mathbf{y}, \mathbf{x}^{\text{obs}}, \mathbf{x}^{\text{miss}}, \mathbf{z}) = p_\phi(\mathbf{y}|\mathbf{x}^{\text{obs}}, \mathbf{x}^{\text{miss}})p_\theta(\mathbf{x}^{\text{obs}}, \mathbf{x}^{\text{miss}}|\mathbf{z})p_\theta(\mathbf{z}). \quad (11)$$

Under the MAR assumption, the relevant substitute of the likelihood is the likelihood of the observed data

$$p_{\phi,\theta}(\mathbf{y}, \mathbf{x}^{\text{obs}}) = \int p_\phi(\mathbf{y}|\mathbf{x}^{\text{obs}}, \mathbf{x}^{\text{miss}})p_\theta(\mathbf{x}^{\text{obs}}, \mathbf{x}^{\text{miss}}|\mathbf{z})p_\theta(\mathbf{z}) \, \mathrm{d}\mathbf{z} \, \mathrm{d}\mathbf{x}^{\text{miss}}. \quad (12)$$

The integral in equation (12) is in all but the simplest cases analytically intractable and direct maximum likelihood methods for learning the parameters $(\theta, \phi)$ cannot be used. In order to learn the parameters of the joint model, we turn to *amortized importance weighted variational inference*, where a lower bound is maximized instead of the log likelihood itself (Burda et al., 2016; Domke & Sheldon, 2018). It is based on the fact that unbiased estimates of a likelihood can be turned into lower bounds of the log-likelihood. Indeed, if $R(\mathbf{x}^{\text{obs}}, \mathbf{y})$ is a random variable such that $\mathbb{E}\left[R(\mathbf{x}^{\text{obs}}, \mathbf{y})\right] = p_{\theta,\phi}(\mathbf{x}^{\text{obs}}, \mathbf{y})$, then $\mathbb{E}\left[\log R(\mathbf{x}^{\text{obs}}, \mathbf{y})\right] \leq \log \mathbb{E}\left[R(\mathbf{x}^{\text{obs}}, \mathbf{y})\right] = \log p_{\theta,\phi}(\mathbf{x}^{\text{obs}}, \mathbf{y})$, which means that $\mathbb{E}\left[\log R(\mathbf{x}^{\text{obs}}, \mathbf{y})\right]$ is a lower bound of the log likelihood. A simple way to find a suitable $R(\mathbf{x}^{\text{obs}}, \mathbf{y})$ is using importance sampling from a variational distribution over $\mathbf{z}$. More specifically, let

$$R_K(\mathbf{x}^{\text{obs}}, \mathbf{y}) = \frac{1}{K} \sum_{k=1}^{K} \frac{p_\phi(\mathbf{y}|\mathbf{x}^{\text{obs}}, \mathbf{x}_k^{\text{miss}})p_\theta(\mathbf{x}^{\text{obs}}|\mathbf{z}_k)p_\theta(\mathbf{z}_k)}{q_\gamma(\mathbf{z}_k|\mathbf{x}^{\text{obs}})}, \quad (13)$$

where $q_{\boldsymbol{\gamma}}(\mathbf{z}|\mathbf{x}^{\text{obs}})$ is the *variational distribution* (learnable proposal) and $(\mathbf{x}_k^{\text{miss}}, \mathbf{z}_k)_{k\in\{1,\ldots,K\}}$ are i.i.d. samples from $p_{\boldsymbol{\theta}}(\mathbf{x}^{\text{miss}}|\mathbf{z})q_{\boldsymbol{\gamma}}(\mathbf{z}|\mathbf{x}^{\text{obs}})$. Then we have that $\mathbb{E}_{(\mathbf{x}_k^{\text{miss}}, \mathbf{z}_k)}\left[R_K(\mathbf{x}^{\text{obs}}, \mathbf{y})\right] = p_{\boldsymbol{\theta},\boldsymbol{\phi}}(\mathbf{x}^{\text{obs}}, \mathbf{y})$ and $\mathbb{E}_{(\mathbf{x}_k^{\text{miss}}, \mathbf{z}_k)}\left[\log R_K(\mathbf{x}^{\text{obs}}, \mathbf{y})\right]$ is a lower bound on the log likelihood. The variational distribution can be parameterized via a neural network with weights $\boldsymbol{\gamma} \in \Gamma$, similarly to Mattei & Frellsen (2019), Nazabal et al. (2020), and Ipsen et al. (2021).

Now that we have a way to bound each per-datum log-likelihood, we can define our *supMIWAE variational bound* $\mathcal{L}_K(\boldsymbol{\theta}, \boldsymbol{\phi}, \boldsymbol{\gamma})$ that is a lower-bound of the log-likelihood of the observed data

$$\mathcal{L}_K(\boldsymbol{\theta}, \boldsymbol{\phi}, \boldsymbol{\gamma}) = \sum_{i=1}^n \mathbb{E}_{(\mathbf{x}_k^{\text{miss}}, \mathbf{z}_k)}\left[\log R_K(\mathbf{x}_i^{\text{obs}}, \mathbf{y}_i)\right] \leq \sum_{i=1}^n \log p_{\boldsymbol{\phi},\boldsymbol{\theta}}(\mathbf{y}_i, \mathbf{x}_i^{\text{obs}}). \tag{14}$$

The lower bound $\mathcal{L}_K(\boldsymbol{\theta}, \boldsymbol{\phi}, \boldsymbol{\gamma})$ is a Monte Carlo objective that can be maximized instead of the log likelihood itself, using techniques from stochastic optimization (see e.g. Mohamed et al., 2020). As detailed in appendix D, it benefits from the usual theoretical advantages of importance-weighted variational bounds: $\mathcal{L}_K(\boldsymbol{\theta}, \boldsymbol{\phi}, \boldsymbol{\gamma})$ converges monotonically to the log-likelihood $\sum_{i=1}^n \log p_{\boldsymbol{\phi},\boldsymbol{\theta}}(\mathbf{y}_i, \mathbf{x}_i^{\text{obs}})$ at speed $1/K$. This means that, the larger the $K$, the closer our objective will be to the true likelihood.

Note that the contribution of any fully observed data point $(\mathbf{x}, \mathbf{y})$ will be simply

$$\log p_{\boldsymbol{\phi}}(\mathbf{y}|\mathbf{x}) + \mathbb{E}_{(\mathbf{z}_k)}\left[\log\left(\frac{1}{K}\sum_{i=1}^K \frac{p_{\boldsymbol{\theta}}(\mathbf{x}|\mathbf{z}_k)p_{\boldsymbol{\theta}}(\mathbf{z}_k)}{q_{\boldsymbol{\gamma}}(\mathbf{z}_k|\mathbf{x})}\right)\right], \tag{15}$$

which is just the sum of the discriminative likelihood and the standard IWAE bound of Burda et al. (2016). An interesting property of our bound is that the discriminative and generative parts can be trained jointly with a single and statistically sound objective. However, if we were to encounter a data set without missing data, equation (15) means that the training of the two parts of the models would be decoupled: the discriminative model would be the same as a standardly trained discriminative model, and the generative model would be the same as a standard IWAE-trained DLVM.

## 2.3 PREDICTION

Once the model has been jointly trained using the supMIWAE bound, it is possible to use it to do prediction with missing data. Indeed *self-normalized importance sampling* can be used to approximate the conditional

$$p_{\boldsymbol{\phi}}(\mathbf{y}|\mathbf{x}^{\text{obs}}) \approx \sum_{i=1}^K w_k p_{\boldsymbol{\phi}}(\mathbf{y}|\mathbf{x}^{\text{obs}}, \mathbf{x}_k^{\text{miss}}), \tag{16}$$

where

$$w_k = \frac{r_k}{r_1 + \ldots + r_K}, \quad \text{and } r_k = \frac{p_{\boldsymbol{\theta}}(\mathbf{x}^{\text{obs}}|\mathbf{z}_k)p_{\boldsymbol{\theta}}(\mathbf{z}_k)}{q_{\boldsymbol{\gamma}}(\mathbf{z}_k|\mathbf{x}^{\text{obs}})}, \tag{17}$$

and $(\mathbf{x}_k^{\text{miss}}, \mathbf{z}_k)_{k\in\{1,\ldots,K\}}$ are i.i.d. samples from $p_{\boldsymbol{\theta}}(\mathbf{x}^{\text{miss}}|\mathbf{z})q_{\boldsymbol{\gamma}}(\mathbf{z}|\mathbf{x}^{\text{obs}})$. Again, this technique mimicks multiple imputation: several imputations are generated using the generative model, are then fed to the classifier, and the predictions of each imputation are averaged.

## 3 RELATED WORK

**Marginalizing over missing covariates** Different approaches have previously been taken to marginalize over missing values in joint models over features and labels. In a series of publications (Ahmad & Tresp, 1993; Tresp et al., 1994; 1995), a closed-form solution using Gaussian Basis Function networks was used to approximate this marginalization, while keeping the neural discriminative model fixed. Ghahramani & Jordan (1995) took a different approach, modeling the joint distribution $p(\mathbf{y}, \mathbf{x})$ directly as a mixture model, obtaining $p(\mathbf{y}|\mathbf{x}^{\text{obs}})$ by conditioning on observed quantities in the joint distribution, thus removing the explicit discriminative neural architecture. In the context of kernel methods, Dick et al. (2008) proposed to learn a distribution over missing values by minimising the regularized empirical risk. Śmieja et al. (2018) trained a Gaussian mixture model (GMM) and discriminative model jointly, and in place of any missing values the activation of the corresponding input neuron was set to the average activation over the GMM conditioned on observed values.

**Consistency of single imputation**   A review of approaches to handling missing data in (non-deep) supervised learning was given by Josse et al. (2019). Here it was shown that under some assumptions about the capacity of the learner, constant imputation is asymptotically consistent in the supervised setting. Le Morvan et al. (2020b) investigated the case of a linear predictor on covariates with missing data, showing that in the presence of missing, the optimal predictor may not be linear and how constant imputation of each feature can be optimized with regards to the model loss. The linear case was further investigated in (Le Morvan et al., 2020a), deriving the analytical form of the optimal predictor and proposing NeuMiss networks to approximate this. Bertsimas et al. (2021) analyzed the impute-then-regress situation and noted that while this paradigm is widely used in practice it incurs a bias in the parameter estimates, but mean imputation is still asymptotically consistent. Le Morvan et al. (2021) showed that the impute-then-regress approach is asymptotically Bayes optimal given a learner with large capacity and that this holds for all missing mechanisms, even *missing not at random* (Rubin, 1976). However, the choice of imputation function affects the difficulty of the subsequent regression task, when not in the asymptotic regime. As a consequence they proposed to learn imputation and regression jointly, where the missing values are handled by a NeuMiss network.

**Other approaches for missing covariates**   Yi et al. (2019) tackled the issue of sparsity, and specifically large variations in sparsity, by introducing sparsity normalization. This addresses the issue of model output covarying with the sparsity level in the input. Ma et al. (2018; 2019) used a permutation invariant setup to avoid imputing missing data in the input of a variational autoencoder. This approach can be readily extended to the supervised setting, using the permutation invariant setup as a modified input layer as we show in appendix A.3. Some tree approaches to regression/classification can use observations with missing values directly (see e.g. Twala et al., 2008; Chen & Guestrin, 2016; Josse et al., 2019; Gómez-Méndez & Joly, 2021). Sparse coding was used by Caiafa et al. (2020) to train a dictionary and a classifier simultaneously.

**DLVMs and missing values**   Deep latent variable models have been applied to missing data problems in the unsupervised setting, focused on inference, single and multiple imputations. Early works assumed that a DLVM had previously been trained on complete data (Rezende et al., 2014; Mattei & Frellsen, 2018), but a variety of techniques to handle incomplete training sets was then proposed, both in M(C)AR (Yoon et al., 2018; Ma et al., 2018; 2019; Ivanov et al., 2019; Mattei & Frellsen, 2019; Li et al., 2019) and MNAR (Ipsen et al., 2021; Ghalebikesabi et al., 2021) cases. In (Li & Marlin, 2020) irregularly sampled time-series are approached as a missing data problem and they propose VAE and GAN based models for handling this. The proposed models can also be used for supervised learning by training a classifier on the latent representations, either by training the model jointly or using a pretrained encoder. Without missing data, the idea of jointly training a VAE and a discriminative model, has been explored by Ji et al. (2020) or Joy et al. (2021).

## 4   EXPERIMENTS

We now evaluate discriminative models trained using the supMIWAE bound on a range of supervised learning tasks. Throughout the experiments the generative part of the model is pretrained using the MIWAE bound of Mattei & Frellsen (2019). The pretrained DLVM can be used to draw single imputations, referred to as MIWAE single imputation, or it can be used as the generative part in the supMIWAE computational structure. Here, the generative part of the model is kept fixed while updating the discriminative part of the model according to the supMIWAE bound. As the DLVM in MIWAE single imputations and the supMIWAE are similar, any difference in performance between the two is attributed to their different strategies for handling missing values. Other strategies for handling missing data, used for comparison in this work, are briefly described in appendix A. Missing values are introduced in the training, validation and test sets according to the missing mechanism used in the given experiment. Predictive performance is measured on a test-set with missing values. In all experiments, there is a computational overhead compared to single imputation of training the supMIWAE that scales linear with the number of importance samples $K$, c.f. equation (13).

### 4.1   2D CLASSIFICATION

A qualitative analysis of the bias in the learnt parameters of the discriminative model, due to single imputation, is set up on simple 2D datasets, see figure 8 of appendix E. The simplicity allows us

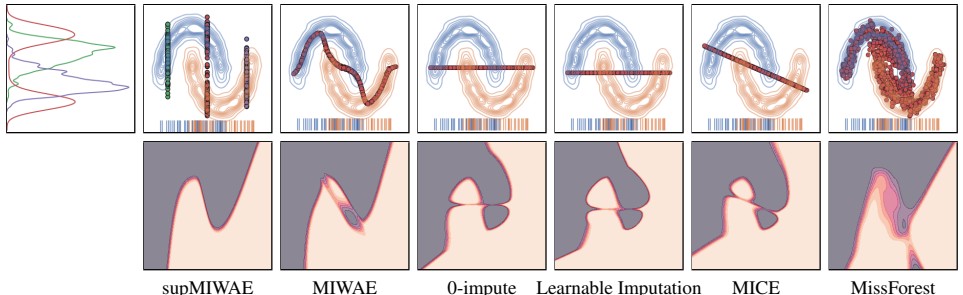

| supMIWAE | MIWAE | 0-impute | Learnable Imputation | MICE | MissForest |

Figure 3: **Top row:** (except top left) kernel density of the half-moons dataset together with imputations from the given model. The supMIWAE does not rely on single imputations, instead it draws multiple importance samples which are passed through the discriminator to give an importance weighted prediction. For the supMIWAE multiple imputations at three different values of the horizontal coordinate are shown and a kernel density of the multiple imputations are shown to the left. For the rest of the methods single imputations are shown in red. **Bottom row:** Decision surfaces learnt, depending on the strategy for handling missing values. The methods based on single imputation need to warp the decision boundaries.

to directly inspect the impact of different imputation strategies on the learnt decision surface and compare this to the marginalization procedure used in the supMIWAE.

All datasets consist of a training set with 3k records and validation and test sets with 1k records. An MCAR missing process is introduced in the horizontal coordinate, where each element becomes missing with probability $m = 0.5$. Further training details are in appendix E. The datasets are such that there will be inherent ambiguity about the class label for some of the records with missing values. For example in the Burger dataset, all records with a missing value will have $p(\mathbf{y} = 1|\mathbf{x}^{\text{obs}}) = 2/3$.

In figure 3 imputations from different imputation models are shown along with the corresponding learnt decision surface. Partially observed examples are shown in the margins of the plots as sticks, colored by their class label. All methods use the same discriminative architecture, only the imputation strategy differs. For the supMIWAE, multiple imputations are shown for three partially observed records, alongside their kernel density. These multiple imputations reflect the appearance of importance samples used during training, for these three partially observed records.

**Decision surface artifacts** The single imputation methods introduce artifacts into the learnt decision surface. As the discriminative model tries to reflect the proper conditional probabilities $p(\mathbf{y}|\mathbf{x}^{\text{obs}})$ at the location of the imputed values, the ambiguity about the class label needs to be reflected in the predictions. Notice how the MIWAE produces nearly optimal imputations in terms of minimizing a mean-square-error and how this translates into decision surface artifacts. These artifacts are related to the notion of *imputation manifolds* used by Le Morvan et al. (2021) to prove their main theorem on the Bayes consitency of impute-then-regress procedures. The remaining datasets are shown in figures 10–13 of appendix E, where also the permutation-invariance setup (appendix A.3) and histogram-based gradient boosting are included. Like the supMIWAE, these two approaches do not rely on explicit imputations.

As the supMIWAE can draw multiple importance samples, instead of one single imputation, the conditional probabilities $p(\mathbf{y}|\mathbf{x}^{\text{obs}})$ can be properly reflected, when averaging over the importance samples. This avoids the single imputation artefacts.

**Predictive performance** Do the artifacts introduced by single imputation methods, compared to marginalizing over the missing values, translate into differences in predictive performance? The predictive performance corresponding to the decision surfaces in figure 3 are shown in table 1 in appendix E. Both the test-set accuracies and log likelihoods are more or less similar across the different strategies for handling missing values. This aligns with the results by Josse et al. (2019); Bertsimas et al. (2021); Le Morvan et al. (2021): the impute-then-regress procedures with powerful learners are consistent and Bayes optimal, i.e. when the learner has high capacity any single imputation can

essentially be undone. While single-imputation methods can lead to Bayes-optimal learners, Le Morvan et al. (2021) address that the difficulty of the learning problem depends on the choice of imputation function. This is seen in figure 4; while the supMIWAE and 0-imputation eventually ends up with similar performance the learning problem appears easier for the supMIWAE. For 0-imputation the learner has to introduce high-frequency artifacts to the decision surface in order to get the conditional probabilities $p(\mathbf{y}|\mathbf{x}^{\text{obs}})$ calibrated, but in the case of a high capacity learner this can eventually be done.

**Limited capacity learner** We now turn to a situation where the capacity of the learner is limited in order to assess how single imputation methods fare against marginalizing over missing values. We start by noting that the decision surface for the Burger dataset, in the complete-data case, can be learnt using two sigmoids. When using MSE-optimal single imputations in the missing-values case this changes, as now 4 sigmoids are needed to make sure the imputations are well calibrated. This illustrates that the capacity of the learner determines whether single imputations can be undone. In figure 5 we explore how the capacity of the learner, in terms of the number of hidden units, affects the predictive performance, depending on the imputation strategy. More capacity experiments are found in figure 14.

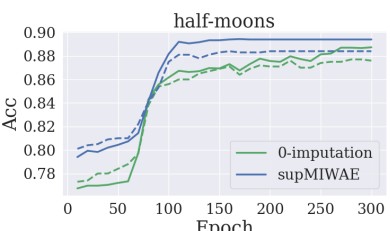

Figure 4: Learning curves on the half-moons dataset: train (*full lines*) and test (*dashed lines*) set accuracies, when the classifier has full capacity.

## 4.2 IMAGE CLASSIFICATION

When the learning problems become harder, strong inductive biases are sometimes used to help learning. This is typically the case with image classification where convolutional neural networks (CNNs) are often used to introduce translational equivariance. The previous section showed that the strategy for handling missing data affects predictive performance when the learner is not in the high capacity regime. This is further investigated in terms of classification accuracy in image experiments with different MCAR missing processes, such as observed squares, missing squares and random dropout, see figures 15–17. The discriminator is a convolutional neural network, see appendix F.1, thus introducing an inductive bias in the learner that has proved useful in image classification in the complete-data case. While CNNs are fit for images, it should be noted that they are not universal approximators, and may consequently have a hard time undoing single imputations.

We apply the supMIWAE and single imputations methods to the MNIST dataset (LeCun et al., 1998) and the fashion MNIST dataset (Xiao et al., 2017) with three different MCAR missing mechanisms: **1)** observed squares, where a $12 \times 12$ randomly placed square is observed, **2)** missing squares, where a $12 \times 12$ randomly placed square is missing, and **3)** dropout, where each pixel is missing with some constant probability $m$, over a range of missing rates.

In practice, we see that this setting where there is a strong inductive bias is similar to our previous low-capacity experiments: for both MNIST and Fashion MNIST, the supMIWAE is significantly

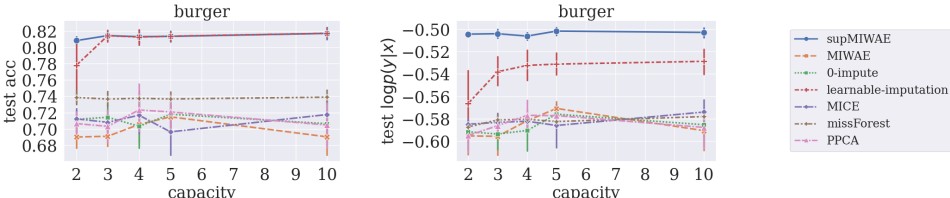

Figure 5: Predictive performance when varying the capacity of the learner in terms of number of hidden units. The discriminative model has the same architecture across methods, only the imputation strategy differs.

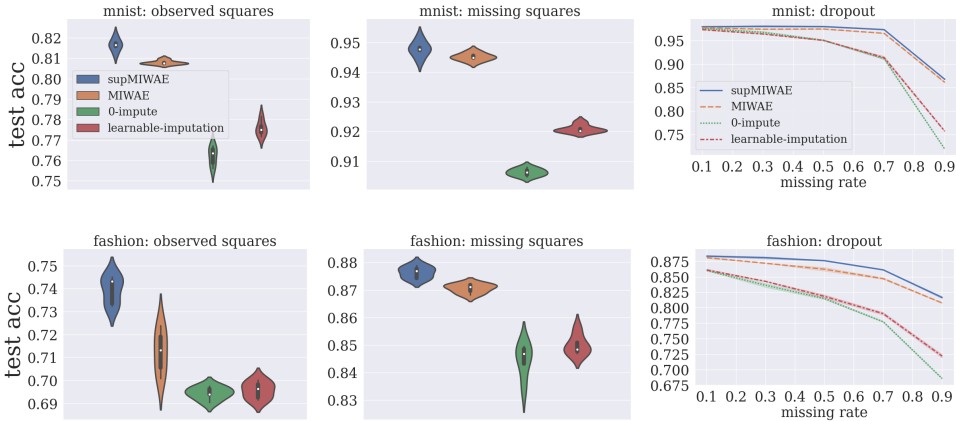

Figure 6: Test set accuracies on the MNIST and Fashion MNIST datasets, with different missing mechanisms. **Left column**: observed squares. **Middle column**: missing squares. **Right column**: random dropout over a range of missing rates.

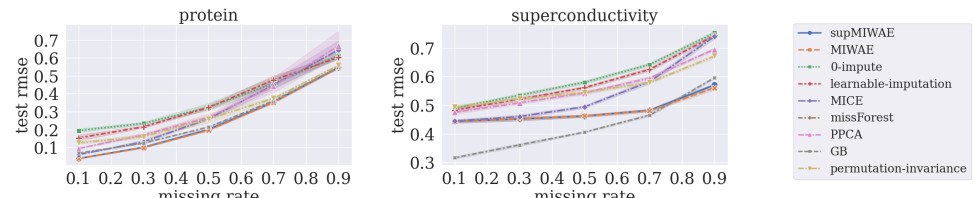

Figure 7: Test-set root mean square error on UCI datasets at varying missing rates.

more accurate than single-imputation competitors, as seen in figure 6. Additional experiments, including experiments on natural images (SVHN) and visualizations of multiple imputations, are available in appendix F.

## 4.3 REGRESSION

We now turn to regression in smaller and lower dimensional datasets from the UCI database (Dua & Graff, 2017). An experimental setup modified from Hernández-Lobato & Adams (2015) and Skafte et al. (2019) is used here. Predictive performance in terms of root-mean-square-error is presented over a range of missing rates. Results are seen in figure 7, and more results can be found in figure 20 of appendix G. There is no clear winner here, although it appears that gradient boosting generally performs quite well, and that accurate single imputations (such as MIWAE's or MissForest's) generally outperform inaccurate ones (such as zero imputation). In general, our joint model performs as well as feeding accurate single imputations.

## 5 CONCLUSION

In the context of supervised deep learning with missing values, single imputation methods are an often used tool. While this approach is asymptotically consistent it incurs a bias in the learnt parameters. We have introduced a scalable approach to marginalizing over missing values using deep generative models, as a probabilistic alternative to single imputation. Besides limiting single imputation artifacts our results indicate that there are two cases where using a generative model can be quite valuable for supervised learning with missing values: when the classifier is not very flexible, or has a strong inductive bias. Moreover, leveraging such a generative model allows us to be able to use untouched any discriminative architecture (for instance, any pretrained one).

REPRODUCIBILITY STATEMENT

Code for reproducing paper experiments is available at `https://github.com/nbip/supMIWAE`.

ACKNOWLEDGEMENTS

This work has been supported by the French government, through the 3IA Côte d'Azur Investments in the Future project managed by the National Research Agency (ANR) with the reference number ANR-19-P3IA-0002. Furthermore, it was supported by the Novo Nordisk Foundation (NNF20OC0062606 and NNF20OC0065611) and the Independent Research Fund Denmark (9131-00082B).

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

## A  TRAINING DEEP SUPERVISED MODELS WITH MISSING DATA

In the following we group different strategies for handling missing data in supervised deep learning.

### A.1  CONSTANT IMPUTATION

We denote imputation methods where there is a single constant for each feature which replaces missing data as *constant imputation*. We define an imputation function $\iota$ in general such that $\iota(\tilde{\mathbf{x}}) \in \mathcal{X}$ and $\iota(\tilde{\mathbf{x}})^{\mathrm{obs}} = \mathbf{x}^{\mathrm{obs}}$. A simple version of constant imputation is 0-imputation, which has the intuitive appeal that the activation from the input node is zeroed out (absent) and no gradient updates are made for the weights for that feature,

$$\iota_0(\tilde{\mathbf{x}}) = \tilde{\mathbf{x}} \odot \mathbf{s} + \mathbf{0} \odot (1 - \mathbf{s}). \tag{18}$$

In the unsupervised setting, constant imputation biases marginal and joint distributions, but Josse et al. (2019); Le Morvan et al. (2021); Bertsimas et al. (2021) have shown that mean imputation can be consistent in the supervised setting. Furthermore, Le Morvan et al. (2020b); Tresp et al. (1994) noted that the constants can be optimized with respect to the prediction loss.

Learnable parameters $\boldsymbol{\lambda} \in \mathcal{X}$ can be inserted in place of the missing data, so that

$$\iota_\lambda(\tilde{\mathbf{x}}) = \tilde{\mathbf{x}} \odot \mathbf{s} + \boldsymbol{\lambda} \odot (1 - \mathbf{s}). \tag{19}$$

We denote this *learnable imputation*.

### A.2  REGRESSIVE/MODEL BASED SINGLE IMPUTATION

In contrast to constant imputation, methods such as MICE (Buuren & Groothuis-Oudshoorn, 2010), MissForest (Stekhoven & Bühlmann, 2012), PPCA (Tipping & Bishop, 1999; Roweis, 1998) or flexible generative models (Mattei & Frellsen, 2019; Yoon et al., 2018) provides imputation of missing features conditional on observed features $p(\mathbf{x}^{\mathrm{miss}}|\mathbf{x}^{\mathrm{obs}})$,

$$\iota(\tilde{\mathbf{x}}) = \tilde{\mathbf{x}} \odot \mathbf{s} + \mathbb{E}_{p(\mathbf{x}^{\mathrm{miss}}|\mathbf{x}^{\mathrm{obs}})}\left[\mathbf{x}\right] \odot (1 - \mathbf{s}) \tag{20}$$

In MICE, this conditional distribution comes from an implicit joint model over the covariates by regressing each covariate with missing values on all the others, in a round-robin fashion. MissForest is somewhat similar to MICE, using random forests as the regressors. Probabilistic PCA defines an explicit joint model over all covariates and single imputations of missing data can be obtained by conditioning on observed covariates. Here, MICE is a proper Bayesian multiple imputation method, while PPCA and MissForest draws multiple imputations without regard to parameter uncertainties. The recent advances in imputation of missing data using deep generative models can also be used here to the extent that they impose a joint distribution over covariates.

### A.3  PERMUTATION INVARIANCE

Ma et al. (2018; 2019) used the permutation invariant setup (Zaheer et al., 2017; Qi et al., 2017) to handle missing data in the context of variational autoencoders. For each feature $j \in \mathrm{obs}(\boldsymbol{s})$, there is an embedding $\boldsymbol{e}_j \in \mathbb{R}^U$, which in general can hold any auxiliary information, but in our case will be a learnable embedding. Each element of the embedding is multiplied with the feature value $\boldsymbol{t}_j = \boldsymbol{e}_j \cdot x_j$ and fed to a neural network $h(\cdot)$, mapping inputs from $\mathbb{R}^U$ to $\mathbb{R}^M$ where $U$ is the embedding dimension and $M$ is the dimension of the code-space being mapped into. This permutation invariant setup finally aggregates all the neural network outputs by a summation to get one fixed-length vector, that can be fed into a discriminative network,

$$\iota_{\mathrm{PI}}(\mathbf{x}^{\mathrm{obs}}) = \sum_{j \in \mathrm{obs}(\boldsymbol{s})} h(\boldsymbol{t}_j). \tag{21}$$

Note that in this case $\iota_{\mathrm{PI}}(\mathbf{x}^{\mathrm{obs}}) \in \mathbb{R}^M$, while in the other imputation functions $\iota(\tilde{\mathbf{x}}) \in \mathcal{X}$.

## A.4 TREE METHODS

Decision trees can in some cases handle missing data directly, without the need for imputation. There are different approaches to handle missing values in tree based methods, see (Josse et al., 2019) for a review. In our experiments we use histogram-based gradient boosting (Friedman, 2001), as implemented in scikit-learn (Pedregosa et al., 2011). This approach uses the Block Propagation method for the missing data. While a discriminative model using MLPs has a fixed capacity, gradient boosting methods can adapt its capacity in terms of tree depth based on a validation set error.

## B GRADIENTS

The log likelihood, when integrating over missing input features, is specified as

$$\log p_{\phi,\theta}(y, \mathbf{x}^{\text{obs}}) = \log p_\phi(y|\mathbf{x}^{\text{obs}}) + \log p_\theta(\mathbf{x}^{\text{obs}}) \tag{22}$$

where

$$\log p_\phi(y|\mathbf{x}^{\text{obs}}) = \log \int p_\phi(y|\mathbf{x}^{\text{obs}}, \mathbf{x}^{\text{miss}}) p_\theta(\mathbf{x}^{\text{miss}}|\mathbf{x}^{\text{obs}}) \, \mathrm{d}\mathbf{x}^{\text{miss}}. \tag{23}$$

Both Tresp et al. (1994) and Ghahramani & Jordan (1995) analyzed the gradient of the log likelihood with respect to the discriminative network parameters,

$$\frac{\partial \log p_{\phi,\theta}(y, \mathbf{x}^{\text{obs}})}{\partial \phi} = \frac{\partial \log p_\phi(y|\mathbf{x}^{\text{obs}})}{\partial \phi} + \frac{\partial \log p_\theta(\mathbf{x}^{\text{obs}})}{\partial \phi} \tag{24}$$

$$= -\frac{1}{p_\phi(y|\mathbf{x}^{\text{obs}})} \frac{\partial p_\phi(y|\mathbf{x}^{\text{obs}})}{\partial \phi} \tag{25}$$

$$= -\frac{1}{p_\phi(y|\mathbf{x}^{\text{obs}})} \frac{\partial}{\partial \phi} \int p_\phi(y|\mathbf{x}^{\text{obs}}, \mathbf{x}^{\text{miss}}) p_\theta(\mathbf{x}^{\text{miss}}|\mathbf{x}^{\text{obs}}) \, \mathrm{d}\mathbf{x}^{\text{miss}} \tag{26}$$

$$= -\frac{1}{p_\phi(y|\mathbf{x}^{\text{obs}})} \int \frac{\partial p_\phi(y|\mathbf{x}^{\text{obs}}, \mathbf{x}^{\text{miss}})}{\partial \phi} p_\theta(\mathbf{x}^{\text{miss}}|\mathbf{x}^{\text{obs}}) \, \mathrm{d}\mathbf{x}^{\text{miss}} \tag{27}$$

Here we see that the gradient of the discriminative network parameters, due to the classification/regression loss, is a weighted sum over all possible missing values, where the missing values are weighted according to their density in the data model, conditional on the observed data.

The gradient with respect to the $\theta$ parameters depends to some extend on the $\phi$ parameters.

$$\frac{\partial \log p_{\phi,\theta}(y, \mathbf{x}^{\text{obs}})}{\partial \theta} = \frac{\partial \log p_\phi(y|\mathbf{x}^{\text{obs}})}{\partial \theta} + \frac{\partial \log p_\theta(\mathbf{x}^{\text{obs}})}{\partial \theta} \tag{28}$$

$$= -\frac{1}{p_\phi(y|\mathbf{x}^{\text{obs}})} \frac{\partial p_\phi(y|\mathbf{x}^{\text{obs}})}{\partial \theta} + \frac{\partial \log p_\theta(\mathbf{x}^{\text{obs}})}{\partial \theta} \tag{29}$$

$$= -\frac{1}{p_\phi(y|\mathbf{x}^{\text{obs}})} \frac{\partial}{\partial \theta} \int p_\phi(y|\mathbf{x}^{\text{obs}}, \mathbf{x}^{\text{miss}}) p_\theta(\mathbf{x}^{\text{miss}}|\mathbf{x}^{\text{obs}}) \, \mathrm{d}\mathbf{x}^{\text{miss}} + \frac{\partial \log p_\theta(\mathbf{x}^{\text{obs}})}{\partial \theta} \tag{30}$$

$$= -\frac{1}{p_\phi(y|\mathbf{x}^{\text{obs}})} \frac{\partial}{\partial \theta} \mathbb{E}_{p_\theta(\mathbf{x}^{\text{miss}}|\mathbf{x}^{\text{obs}})} \left[ p_\phi(y|\mathbf{x}^{\text{obs}}, \mathbf{x}^{\text{miss}}) \right] + \frac{\partial \log p_\theta(\mathbf{x}^{\text{obs}})}{\partial \theta} \tag{31}$$

$$\tag{32}$$

There are contributions to the gradient from both the generative part of the model and the discriminative part. This means that, if needed, the discriminative model can affect the generative model, as is the case in (Ipsen et al., 2021). The gradient of the expectation can be approached using the reparameterization trick (Kingma & Welling, 2014; Rezende et al., 2014).

## C RELATION TO MULTIPLE IMPUTATION

The gold standard for imputation is *multiple imputation* (Rubin, 1996). This requires learning a model of the observed covariates $\mathbf{x}^{\text{obs}}$ with parameters $\theta$. $K$ sets of imputations are then drawn from

the posterior predictive distribution

$$\mathbf{x}_k^{\text{miss}} \sim p(\mathbf{x}^{\text{miss}}|\mathbf{x}^{\text{obs}}), \ k = 1, \ldots, K \tag{33}$$

where $p(\mathbf{x}^{\text{miss}}|\mathbf{x}^{\text{obs}}) = \int p(\mathbf{x}^{\text{miss}}|\boldsymbol{\theta})p(\boldsymbol{\theta}|\mathbf{x}^{\text{obs}})\,\mathrm{d}\boldsymbol{\theta}$ if the model is Bayesian (see e.g. Gelman et al., 2013, chapter 18.2). Several frequentist multiple imputation methods also exist, where the uncertainty with respect to the imputation model parameters is disregarded. In that case, the Bayesian posterior predictive is replaced by a conditional distribution $p(\mathbf{x}^{\text{miss}}|\mathbf{x}^{\text{obs}}) = p_{\hat{\boldsymbol{\theta}}}(\mathbf{x}^{\text{miss}}|\mathbf{x}^{\text{obs}})$, where $\hat{\boldsymbol{\theta}}$ is any point estimate of $\boldsymbol{\theta}$.

In the case of regression/classification, if we roughly follow the notations of our paper, the analysis proceeds as follows

- For each set of imputed values $(\mathbf{x}_{ik}^{\text{miss}})_{i \leq n, k \leq K}$, use a learning algorithm to estimate the discriminative parameters $\boldsymbol{\phi}_1, \ldots, \boldsymbol{\phi}_K$:

$$\hat{\boldsymbol{\phi}}_1 = \arg\max_{\boldsymbol{\phi}} \sum_{i=1}^{n} \log p_{\boldsymbol{\phi}}(y_i|\mathbf{x}_i^{\text{obs}}, \mathbf{x}_{i1}^{\text{miss}}) \tag{34}$$

$$\vdots$$

$$\hat{\boldsymbol{\phi}}_K = \arg\max_{\boldsymbol{\phi}} \sum_{i=1}^{n} \log p_{\boldsymbol{\phi}}(y_i|\mathbf{x}_i^{\text{obs}}, \mathbf{x}_{iK}^{\text{miss}}) \tag{35}$$

- Obtain an average estimate of $\phi$ using

$$\bar{\boldsymbol{\phi}}_K = \frac{1}{K} \sum_{k=1}^{K} \hat{\boldsymbol{\phi}}_k \tag{36}$$

and a variance estimate can similarly be obtained as described in (Gelman et al., 2013; Rubin, 1996).

In the limit of infinite imputations, we can write this as the conditional expectation

$$\bar{\boldsymbol{\phi}} = \mathbb{E}_{\mathbf{x}_1^{\text{miss}}, \ldots, \mathbf{x}_n^{\text{miss}}} \left[ \arg\max_{\boldsymbol{\phi}} \sum_{i=1}^{n} \log p_{\boldsymbol{\phi}}(y_i|\mathbf{x}_i^{\text{obs}}, \mathbf{x}_i^{\text{miss}}) \,\middle|\, \mathbf{x}_1^{\text{obs}}, \ldots, \mathbf{x}_n^{\text{obs}} \right]. \tag{37}$$

Instead of fitting $K$ conditional models to $K$ imputed datasets, our approach (when $\boldsymbol{\theta}$ is fixed) is, in comparison, to directly maximize the expectation of the conditional distribution with respect to the missing data

$$\bar{\boldsymbol{\phi}} = \arg\max_{\boldsymbol{\phi}} \sum_{i=1}^{n} \log \mathbb{E}_{\mathbf{x}_i^{\text{miss}}} \left[ p_{\boldsymbol{\phi}}(y_i|\mathbf{x}_i^{\text{obs}}, \mathbf{x}_i^{\text{miss}})|\mathbf{x}_i^{\text{obs}} \right]. \tag{38}$$

Also note that we have no need for several fixed sets of imputations, at each gradient step a new set of samples are drawn from the imputation model, $\mathbf{x}^{\text{miss}} \sim p_{\boldsymbol{\theta}}(\mathbf{x}^{\text{miss}}|\mathbf{x}^{\text{obs}})$. The number of imputations can be set dynamically, using $\sim$ 5–50 during training and $\sim$ 10k for testing. Our approach, using DLVMs as imputation models, is not Bayesian, but any available proper Bayesian imputation model could be used in place of the DLVMs.

In practice, an important difference between traditional multiple imputation and our approach, is that multiple imputation uses a uniform weight $1/K$ for each imputation, while we use importance weights, that will treat each imputation based on its likelihood.

## D    THEORETICAL PROPERTIES OF THE SUPMIWAE BOUND

We will briefly see how the supMIWAE bound has the same theoretical properties as IWAE bounds. We follow Ipsen et al. (2021, appendix D), who were following Burda et al. (2016) and Domke & Sheldon (2018). Recall the definition of the bound: we have

$$R_K(\mathbf{x}^{\text{obs}}, \mathbf{y}) = \frac{1}{K} \sum_{k=1}^{K} \frac{p_{\boldsymbol{\phi}}(\mathbf{y}|\mathbf{x}^{\text{obs}}, \mathbf{x}_k^{\text{miss}})p_{\boldsymbol{\theta}}(\mathbf{x}^{\text{obs}}|\mathbf{z}_k)p_{\boldsymbol{\theta}}(\mathbf{z}_k)}{q_{\boldsymbol{\gamma}}(\mathbf{z}_k|\mathbf{x}^{\text{obs}})}, \tag{39}$$

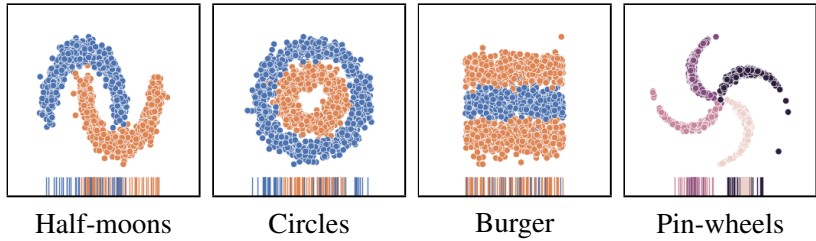

Figure 8: 2D datasets with two or more classes.

and

$$\mathcal{L}_K(\boldsymbol{\theta}, \boldsymbol{\phi}, \boldsymbol{\gamma}) = \sum_{i=1}^{n} \mathbb{E}_{(\mathbf{x}_k^{\text{miss}}, \mathbf{z}_k)} \left[ \log R_K(\mathbf{x}_i^{\text{obs}}, \mathbf{y}_i) \right]. \tag{40}$$

The $i$th term of the sum can be seen as a regular IWAE bound with observation $(\mathbf{x}^{\text{obs}}, \mathbf{y})$, latent variable $(\mathbf{z}, \mathbf{x}^{\text{miss}})$ with prior $p_{\boldsymbol{\theta}}(\mathbf{x}^{\text{miss}}|\mathbf{z})p_{\boldsymbol{\theta}}(\mathbf{z})$, and variational distribution $p_{\boldsymbol{\theta}}(\mathbf{x}^{\text{miss}}|\mathbf{z})q_{\boldsymbol{\gamma}}(\mathbf{z}|\mathbf{x}^{\text{obs}})$. Therefore, Theorem 1 from Burda et al. (2016) can be applied, and leads to the monotonicity of the bound: $\mathcal{L}_1(\boldsymbol{\theta}, \boldsymbol{\phi}, \boldsymbol{\gamma}) \leq \ldots \leq \mathcal{L}_K(\boldsymbol{\theta}, \boldsymbol{\phi}, \boldsymbol{\gamma})$.

To show that the bound additionally converges to the log-likelihood when $K \longrightarrow \infty$, we use Theorem 3 of Domke & Sheldon (2018) for all $n$ terms of the sum that constitutes the bound.

**Theorem.** *Assuming that, for all $i \in \{1, ..., n\}$,*

- *there exists $\alpha_i > 0$ such that $\mathbb{E} \left[ |R_1(\mathbf{x}_i^{obs}, \mathbf{y}_i) - p_{\boldsymbol{\theta}, \boldsymbol{\phi}}(\mathbf{x}_i^{obs}, \mathbf{y}_i)|^{2+\alpha_i} \right] < \infty,$*

- $\limsup_{K \longrightarrow \infty} \mathbb{E} \left[ 1/R_K \right] < \infty,$

*the supMIWAE bound converges to the true joint likelihood at rate $1/K$:*

$$\sum_{i=1}^{n} \log p_{\boldsymbol{\phi}, \boldsymbol{\theta}}(\mathbf{y}_i, \mathbf{x}_i^{obs}) - \mathcal{L}_K(\boldsymbol{\theta}, \boldsymbol{\phi}, \boldsymbol{\gamma}) \underset{K \to \infty}{\sim} \frac{1}{K} \sum_{i=1}^{n} \frac{\text{Var}[R_1(\mathbf{x}_i^{obs}, \mathbf{y}_i)]}{2p_{\boldsymbol{\theta}, \boldsymbol{\phi}}(\mathbf{x}_i^{obs}, \mathbf{y}_i)^2}. \tag{41}$$

## E 2D CLASSIFICATION

The datasets used in the 2D classification experiments are shown in figure 8. The Half-moons and Circles datasets are from scikit-learn (Pedregosa et al., 2011). The Pin-wheel dataset is modified from Johnson et al. (2016)[1]. The Burger dataset, we have designed to illustrate the case where optimal single imputation requires severe changes in the decision surface.

The discriminative model is an MLP with 3 hidden layers with 50 hidden units each. The final layer parameterizes a categorical distribution over the dataset classes. The generative model used for the MIWAE single imputations and the supMIWAE consists of an encoder and decoder with 3 hidden layers containing 50 hidden units each. The encoder parameterizes an isotropic Gaussian distribution over a latent space with dimension $p$, same as the feature space. The decoder parameterizes an isotropic Gaussian distribution over the feature space. In figure 9 samples from the generative model are shown, that is, samples from the MIWAE trained on each of the datasets.

In (Śmieja et al., 2018) a Gaussian Mixture Model, GMM, is used to model the density of the covariates while jointly training a discriminator. In case of missing values, the discriminator's first input neuron's activations are set to the average activations as found over a conditional distribution over the missing values in the GMM. We adapt the official code [2] to this experiment by adjusting the number of hidden layers and number of mixture components while keeping the original training setup as is.

---

[1]https://github.com/mattjj/svae
[2]https://github.com/lstruski/Processing-of-missing-data-by-neural-networks

In figures 10–13 decision surfaces and imputations are shown. The corresponding quantitative results are in tables 1–4.

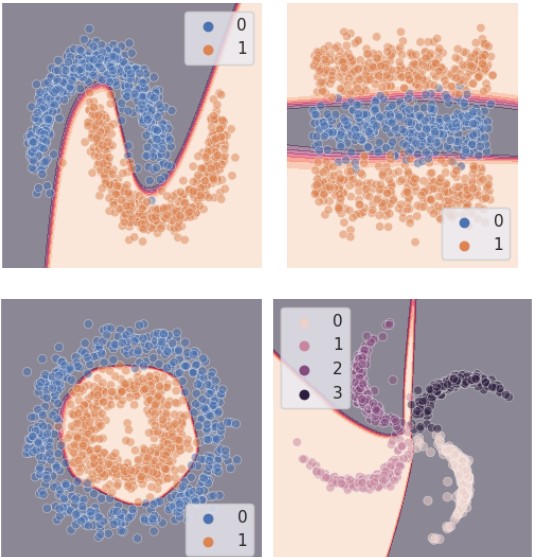

Figure 9: Samples from the MIWAE, fitted to the data with missing values on the four 2D datasets. The class color is determined by afterwards sampling from the discriminative distribution.

Table 1: Half-moons test-set results

| model | test acc | - on missing | - on observed | test log $p(y\|x)$ | - on missing | - on observed |
|---|---|---|---|---|---|---|
| supMIWAE | $0.88 \pm 0.01$ | $0.77 \pm 0.01$ | $1.00 \pm 0.00$ | $-0.42 \pm 0.00$ | $-0.52 \pm 0.01$ | $-0.32 \pm 0.00$ |
| MIWAE | $0.88 \pm 0.00$ | $0.77 \pm 0.01$ | $0.99 \pm 0.00$ | $-0.42 \pm 0.01$ | $-0.52 \pm 0.01$ | $-0.33 \pm 0.01$ |
| 0-impute | $0.88 \pm 0.01$ | $0.77 \pm 0.02$ | $1.00 \pm 0.00$ | $-0.42 \pm 0.00$ | $-0.52 \pm 0.01$ | $-0.32 \pm 0.00$ |
| learnable-imputation | $0.89 \pm 0.01$ | $0.78 \pm 0.01$ | $1.00 \pm 0.00$ | $-0.42 \pm 0.00$ | $-0.52 \pm 0.01$ | $-0.32 \pm 0.00$ |
| MICE | $0.89 \pm 0.01$ | $0.77 \pm 0.01$ | $1.00 \pm 0.00$ | $-0.42 \pm 0.00$ | $-0.52 \pm 0.01$ | $-0.32 \pm 0.00$ |
| missForest | $0.87 \pm 0.00$ | $0.75 \pm 0.01$ | $0.99 \pm 0.01$ | $-0.45 \pm 0.00$ | $-0.53 \pm 0.01$ | $-0.37 \pm 0.00$ |
| PPCA | $0.88 \pm 0.01$ | $0.77 \pm 0.02$ | $1.00 \pm 0.00$ | $-0.42 \pm 0.00$ | $-0.52 \pm 0.01$ | $-0.32 \pm 0.00$ |
| GB | $0.88 \pm 0.01$ | $0.76 \pm 0.02$ | $1.00 \pm 0.00$ | $-0.42 \pm 0.00$ | $-0.52 \pm 0.01$ | $-0.32 \pm 0.00$ |
| permutation-invariance | $0.88 \pm 0.01$ | $0.77 \pm 0.01$ | $1.00 \pm 0.00$ | $-0.42 \pm 0.00$ | $-0.52 \pm 0.01$ | $-0.32 \pm 0.01$ |
| (Śmieja et al., 2018) | $0.83 \pm 0.04$ | | | | | |

Table 2: Burger test-set results

| model | test acc | - on missing | - on observed | test log $p(y\|x)$ | - on missing | - on observed |
|---|---|---|---|---|---|---|
| supMIWAE | $0.81 \pm 0.01$ | $0.64 \pm 0.03$ | $0.97 \pm 0.01$ | $-0.51 \pm 0.00$ | $-0.66 \pm 0.01$ | $-0.35 \pm 0.01$ |
| MIWAE | $0.78 \pm 0.01$ | $0.63 \pm 0.03$ | $0.93 \pm 0.02$ | $-0.52 \pm 0.01$ | $-0.67 \pm 0.01$ | $-0.38 \pm 0.01$ |
| 0-impute | $0.80 \pm 0.01$ | $0.64 \pm 0.03$ | $0.96 \pm 0.01$ | $-0.51 \pm 0.00$ | $-0.66 \pm 0.01$ | $-0.35 \pm 0.01$ |
| learnable-imputation | $0.81 \pm 0.01$ | $0.64 \pm 0.03$ | $0.97 \pm 0.01$ | $-0.51 \pm 0.00$ | $-0.66 \pm 0.01$ | $-0.35 \pm 0.01$ |
| MICE | $0.80 \pm 0.01$ | $0.64 \pm 0.03$ | $0.96 \pm 0.01$ | $-0.51 \pm 0.00$ | $-0.66 \pm 0.01$ | $-0.35 \pm 0.01$ |
| missForest | $0.73 \pm 0.02$ | $0.54 \pm 0.03$ | $0.93 \pm 0.02$ | $-0.58 \pm 0.01$ | $-0.69 \pm 0.02$ | $-0.47 \pm 0.02$ |
| PPCA | $0.80 \pm 0.01$ | $0.64 \pm 0.03$ | $0.96 \pm 0.01$ | $-0.51 \pm 0.01$ | $-0.66 \pm 0.01$ | $-0.35 \pm 0.01$ |
| GB | $0.80 \pm 0.01$ | $0.63 \pm 0.03$ | $0.97 \pm 0.01$ | $-0.51 \pm 0.01$ | $-0.66 \pm 0.01$ | $-0.35 \pm 0.01$ |
| permutation-invariance | $0.81 \pm 0.01$ | $0.64 \pm 0.03$ | $0.97 \pm 0.01$ | $-0.51 \pm 0.00$ | $-0.66 \pm 0.01$ | $-0.35 \pm 0.01$ |
| (Śmieja et al., 2018) | $0.79 \pm 0.06$ | | | | | |

Table 3: Circles test-set results

| model | test acc | - on missing | - on observed | test log $p(y\|x)$ | - on missing | - on observed |
|---|---|---|---|---|---|---|
| supMIWAE | $0.88 \pm 0.01$ | $0.77 \pm 0.01$ | $0.99 \pm 0.01$ | $-0.44 \pm 0.01$ | $-0.56 \pm 0.01$ | $-0.32 \pm 0.01$ |
| MIWAE | $0.87 \pm 0.01$ | $0.76 \pm 0.02$ | $0.99 \pm 0.01$ | $-0.45 \pm 0.01$ | $-0.56 \pm 0.01$ | $-0.33 \pm 0.01$ |
| 0-impute | $0.88 \pm 0.01$ | $0.77 \pm 0.02$ | $0.99 \pm 0.00$ | $-0.44 \pm 0.01$ | $-0.56 \pm 0.00$ | $-0.33 \pm 0.00$ |
| learnable-imputation | $0.88 \pm 0.01$ | $0.77 \pm 0.02$ | $0.99 \pm 0.01$ | $-0.44 \pm 0.00$ | $-0.55 \pm 0.01$ | $-0.33 \pm 0.01$ |
| MICE | $0.88 \pm 0.01$ | $0.77 \pm 0.02$ | $0.99 \pm 0.01$ | $-0.44 \pm 0.01$ | $-0.56 \pm 0.01$ | $-0.33 \pm 0.00$ |
| missForest | $0.86 \pm 0.01$ | $0.75 \pm 0.01$ | $0.98 \pm 0.01$ | $-0.48 \pm 0.01$ | $-0.56 \pm 0.01$ | $-0.39 \pm 0.01$ |
| PPCA | $0.88 \pm 0.01$ | $0.77 \pm 0.02$ | $0.99 \pm 0.00$ | $-0.44 \pm 0.00$ | $-0.56 \pm 0.01$ | $-0.33 \pm 0.00$ |
| GB | $0.87 \pm 0.01$ | $0.76 \pm 0.01$ | $0.99 \pm 0.00$ | $-0.44 \pm 0.01$ | $-0.56 \pm 0.01$ | $-0.33 \pm 0.00$ |
| permutation-invariance | $0.88 \pm 0.01$ | $0.77 \pm 0.02$ | $0.99 \pm 0.00$ | $-0.44 \pm 0.00$ | $-0.56 \pm 0.01$ | $-0.32 \pm 0.00$ |
| (Śmieja et al., 2018) | $0.86 \pm 0.02$ | | | | | |

Table 4: Pin-wheel test-set results

| model | test acc | - on missing | - on observed | test log $p(y\|x)$ | - on missing | - on observed |
|---|---|---|---|---|---|---|
| supMIWAE | $0.88 \pm 0.01$ | $0.77 \pm 0.01$ | $1.00 \pm 0.00$ | $-0.89 \pm 0.01$ | $-1.02 \pm 0.01$ | $-0.75 \pm 0.00$ |
| MIWAE | $0.88 \pm 0.01$ | $0.76 \pm 0.01$ | $1.00 \pm 0.00$ | $-0.88 \pm 0.01$ | $-1.01 \pm 0.01$ | $-0.75 \pm 0.00$ |
| 0-impute | $0.88 \pm 0.01$ | $0.77 \pm 0.01$ | $1.00 \pm 0.00$ | $-0.88 \pm 0.01$ | $-1.02 \pm 0.00$ | $-0.75 \pm 0.00$ |
| learnable-imputation | $0.88 \pm 0.00$ | $0.77 \pm 0.01$ | $1.00 \pm 0.00$ | $-0.88 \pm 0.01$ | $-1.02 \pm 0.01$ | $-0.75 \pm 0.00$ |
| MICE | $0.88 \pm 0.01$ | $0.77 \pm 0.01$ | $1.00 \pm 0.00$ | $-0.88 \pm 0.01$ | $-1.02 \pm 0.01$ | $-0.75 \pm 0.00$ |
| missForest | $0.87 \pm 0.00$ | $0.75 \pm 0.00$ | $1.00 \pm 0.00$ | $-0.90 \pm 0.01$ | $-1.01 \pm 0.00$ | $-0.80 \pm 0.00$ |
| PPCA | $0.88 \pm 0.01$ | $0.77 \pm 0.01$ | $1.00 \pm 0.00$ | $-0.88 \pm 0.01$ | $-1.02 \pm 0.01$ | $-0.75 \pm 0.00$ |
| GB | $0.87 \pm 0.01$ | $0.74 \pm 0.02$ | $1.00 \pm 0.00$ | $-0.88 \pm 0.01$ | $-1.02 \pm 0.01$ | $-0.75 \pm 0.00$ |
| permutation-invariance | $0.88 \pm 0.01$ | $0.77 \pm 0.01$ | $1.00 \pm 0.00$ | $-0.88 \pm 0.01$ | $-1.02 \pm 0.01$ | $-0.75 \pm 0.00$ |

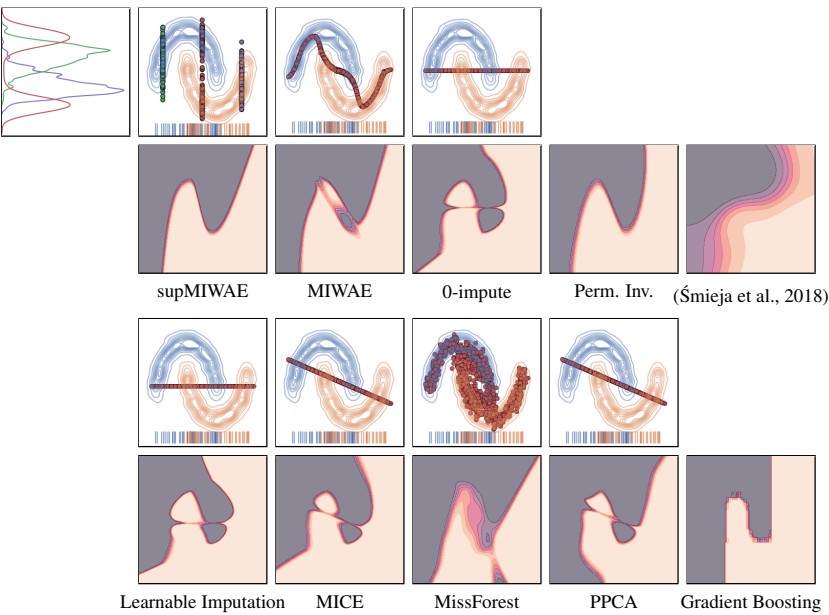

Figure 10: **First and third rows:** (except top left) kernel density of the half-moons dataset together with imputations from the given model. The supMIWAE does not rely on single imputations, instead it draws multiple importance samples which are passed through the discriminator to give an importance weighted prediction. For the supMIWAE multiple imputations at three different values of the horizontal coordinate are shown and a kernel density of the multiple imputations are shown to the left. Permutation invariance, the joint GMM and discriminator approach of Śmieja et al. (2018) and gradient boosting does not rely on explicit imputations, for the rest of the methods single imputations are shown in red. **Second and fourth rows:** Decision surfaces learnt, depending on the strategy for handling missing values.

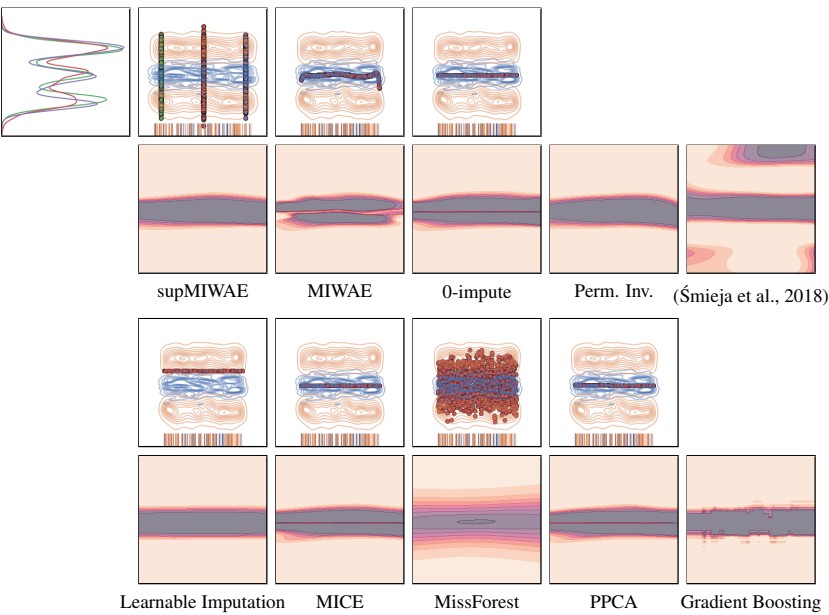

Figure 11: **First and third rows:** (except top left) kernel density of the burger dataset together with imputations from the given model. The supMIWAE does not rely on single imputations, instead it draws multiple importance samples which are passed through the discriminator to give an importance weighted prediction. For the supMIWAE multiple imputations at three different values of the horizontal coordinate are shown and a kernel density of the multiple imputations are shown to the left. Permutation invariance, the joint GMM and discriminator approach of Śmieja et al. (2018) and gradient boosting does not rely on explicit imputations, for the rest of the methods single imputations are shown in red. **Second and fourth rows:** Decision surfaces learnt, depending on the strategy for handling missing values.

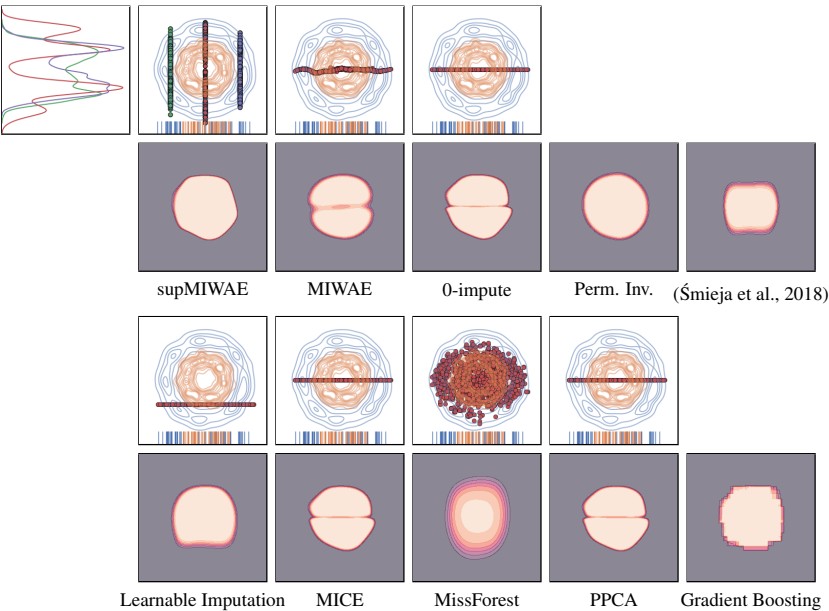

Figure 12: **First and third rows:** (except top left) kernel density of the circles dataset together with imputations from the given model. The supMIWAE does not rely on single imputations, instead it draws multiple importance samples which are passed through the discriminator to give an importance weighted prediction. For the supMIWAE multiple imputations at three different values of the horizontal coordinate are shown and a kernel density of the multiple imputations are shown to the left. Permutation invariance, the joint GMM and discriminator approach of Śmieja et al. (2018) and gradient boosting does not rely on explicit imputations, for the rest of the methods single imputations are shown in red. **Second and fourth rows:** Decision surfaces learnt, depending on the strategy for handling missing values.

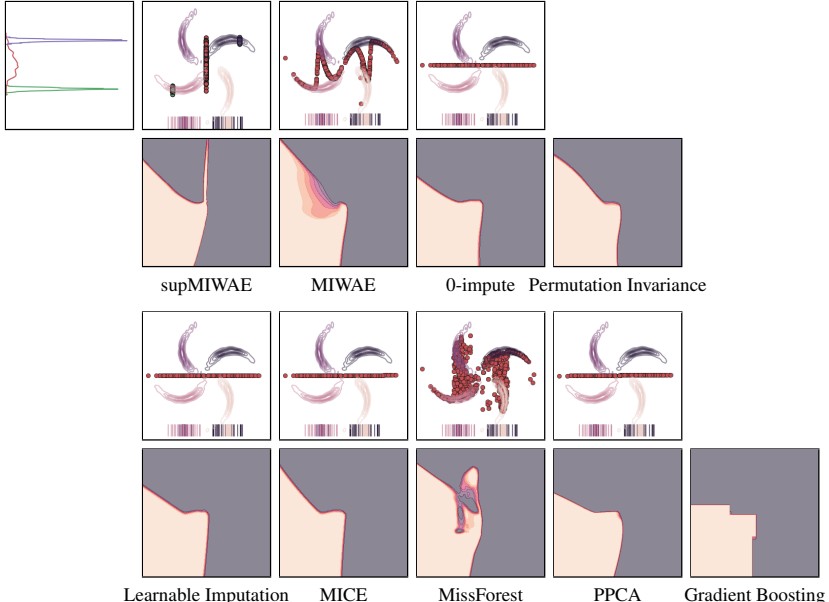

Figure 13: **First and third rows:** (except top left) kernel density of the pin-wheel dataset together with imputations from the given model. The supMIWAE does not rely on single imputations, instead it draws multiple importance samples which are passed through the discriminator to give an importance weighted prediction. For the supMIWAE multiple imputations at three different values of the horizontal coordinate are shown and a kernel density of the multiple imputations are shown to the left. Permutation invariance and gradient boosting does not rely on explicit imputations, for the rest of the methods single imputations are shown in red. **Second and fourth rows:** Decision surfaces learnt, depending on the strategy for handling missing values.

## E.1 CAPACITY

The predictive performance of the discriminator at different capacities, in terms of number of hidden units, is shown in figure 14.

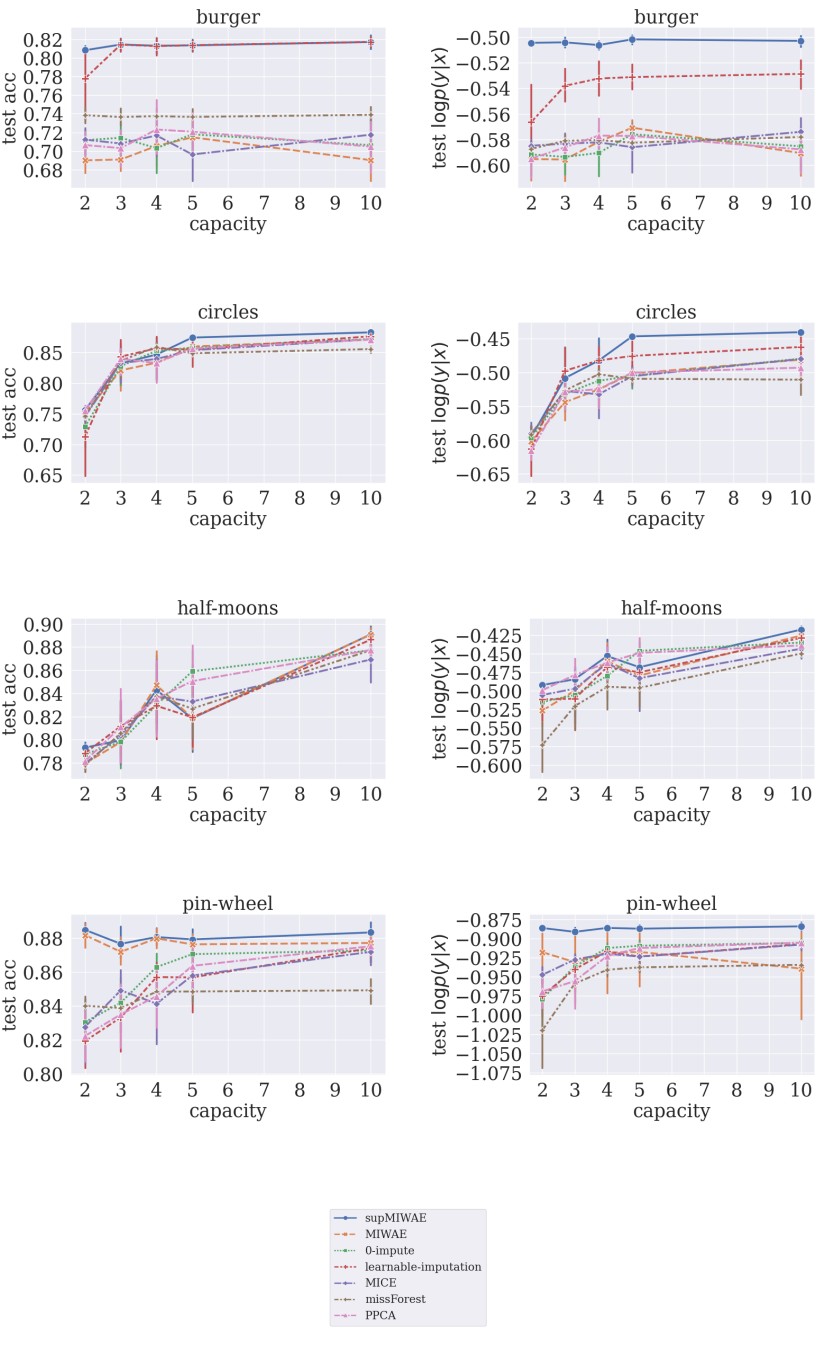

Figure 14: Predictive performance when varying the capacity of the learner, in terms of hidden units.

# F   IMAGE CLASSIFICATION

## F.1   TRAINING DETAILS FOR MNIST/FMNIST

For all MNIST/fMNIST experiments, the discriminative model is a convolutional neural network with four layers and a categorical distribution over classes, see table 5. The generative model uses a convolutional architecture similar to the DCGAN (Radford et al., 2015) and a latent space of dimension 128. A Continuous Bernoulli (Loaiza-Ganem & Cunningham, 2019) observation model is used for MNIST and the discretized logistic distribution (Salimans et al., 2017) is used as the observation model for Fashion MNIST. During training, $K = 5$ importance samples and a batch size of 128 are used.

Table 5: Discriminative network for MNIST/fMNIST classification.

| Action (resulting layer size) |
| --- |
| Input $x$ ($28 \times 28 \times 1$) |
| Conv2D($14 \times 14 \times 16$) |
| Conv2D($7 \times 7 \times 32$) |
| Conv2D($3 \times 3 \times 64$) |
| Reshape(576) |
| Class probabilities: Dense(10) |

## F.2   IMPUTATIONS

Figures 15–17 show imputations on the two image datasets for the three different MCAR missing processes. The first column contains data with missing values. The second column is single imputation by the MIWAE, and following columns are MIWAE multiple imputations using sampling-importance-resampling.

## F.3   NATURAL IMAGES (SVHN)

In order to assess classification of natural images we now turn to the Street View House Numbers dataset (SVHN, Netzer et al., 2011), with different missing mechanisms. In the "observed squares" missing mechanism $20 \times 20$ randomly placed squares are observed while in the "missing squares" missing mechanism $16 \times 16$ randomly placed squares are missing. In the "random dropout" missing mechanism pixels are missing with probability $m = 0.5$.

In the "observed squares" experiment we apply both a low capacity 4 layer discriminator (1 repetition) and a higher capacity 8 layer discriminator (3 repetitions) using the convolutional gated blocks from Dauphin et al. (2017), see architectures in tables 7 and 8. In table 6, test set accuracies are shown and for the 8 layer discriminator, the accuracy on the fully observed test set is also shown. In the low capacity regime, the supMIWAE outperforms the two single imputation methods and performs on par with MIWAE. With the higher capacity discriminator supMIWAE has performance similar to 0-imputation and learnable imputation, while the MIWAE imputation is slightly worse. This displays the same phenomenon as in the 2D classification experiments, where powerful learners are able to undo single imputations. While the discriminative models learnt based on constant imputation can achieve the same performance as the supMIWAE, the kinds of discriminative models that are learnt are different. In this case, when applied to a fully observed test-set the discriminator learnt by the supMIWAE outperforms the other models.

The high capacity discriminator is also applied to the "missing squares" and "random dropout" missing mechanisms, see figure 18 for the test-set accuracies (3 repetitions). In figure 19 the models trained on data with missing values are applied to a fully observed test-set.

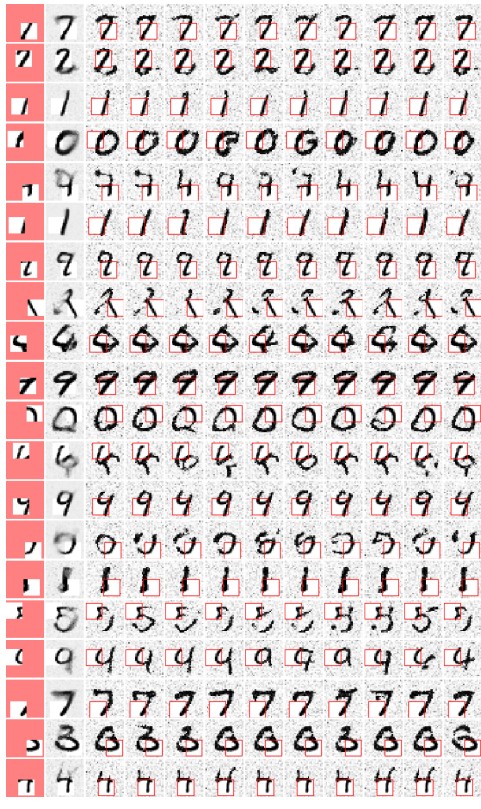 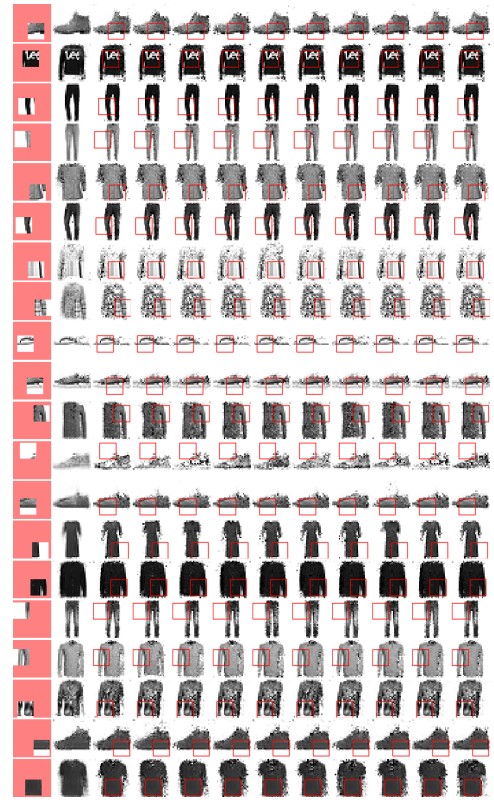

(a) MNIST, Continuous Bernoulli.      (b) Fashion, discretized logistic.

Figure 15: Observed squares. Left column contains data with missing values represented as red pixels. Second column contains single imputations from the MWIAE using *self-normalized importance sampling*. Following columns contain multiple from the MIWAE using *sampling-importance-resampling*.

Table 6: SVHN observed squares classification accuracies

| model | acc (incomplete test set) | | acc (complete test set) |
| | 4 layers | 8 layers | 8 layers |
| --- | --- | --- | --- |
| supMIWAE | 0.8264 | $0.8795 \pm 0.0027$ | $0.9268 \pm 0.0015$ |
| MIWAE | 0.8233 | $0.8745 \pm 0.0024$ | $0.9209 \pm 0.0010$ |
| 0-impute | 0.8155 | $0.8819 \pm 0.0024$ | $0.9059 \pm 0.0064$ |
| learnable-imputation | 0.8139 | $0.8798 \pm 0.0042$ | $0.8953 \pm 0.0234$ |

## G  REGRESSION

All discriminative models are neural networks with one hidden layer and 50 hidden units, where the final layer of the neural networks parameterizes a Gaussian distribution. The datasets are split randomly 20 times with 90% of the data in a training set and 10% in a test set. A validation set with 10% of the training data is used for early stopping and a batch size of 256 is used. For the permutation invariant setup, an embedding of dimension $U = 20$ and fixed length vectors $\iota_{\mathrm{PI}}(\mathbf{x}^{\mathrm{obs}})$ of dimension $M = 10$ are used. Missing values are introduced *completely at random* in the training, validation and test sets by removing each feature with probability $m$, where $m$ is the missing rate. In order to avoid reducing the sample size, due to completely missing covariates, whenever all the covariates of an observation go missing, we set one of them to observed, selected at random.

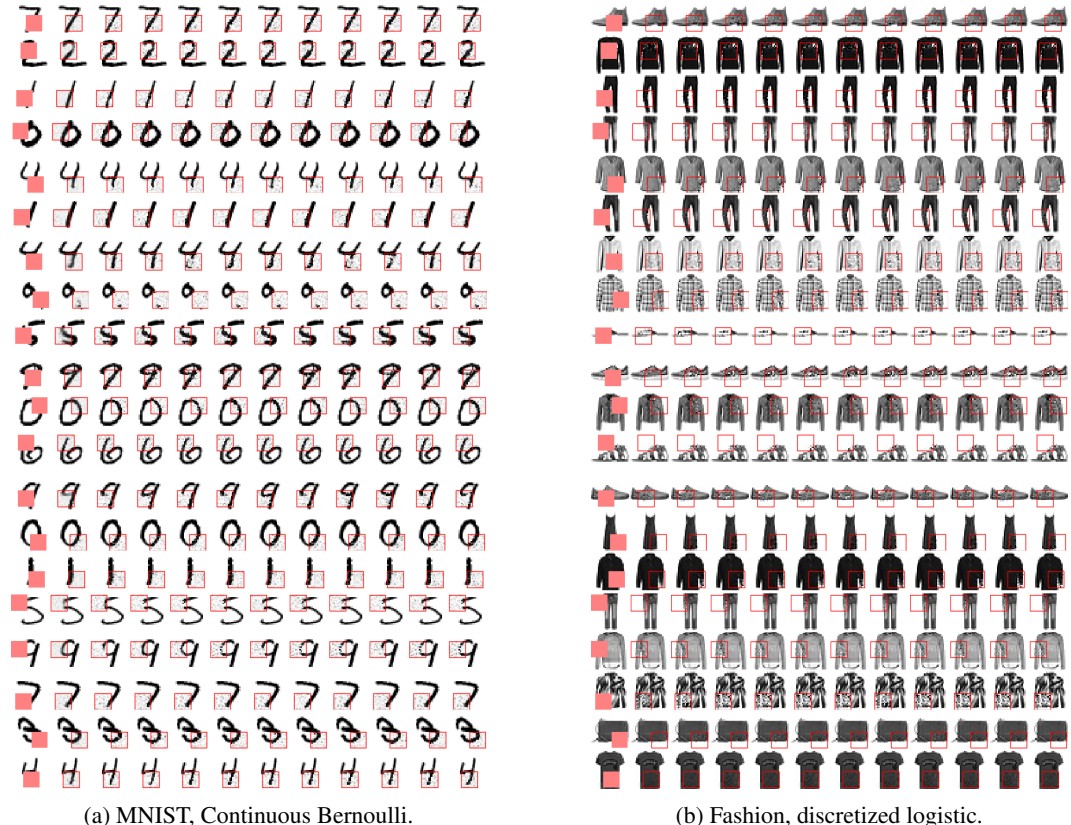

(a) MNIST, Continuous Bernoulli.          (b) Fashion, discretized logistic.

Figure 16: Missing squares. Left column contains data with missing values represented as red pixels. Second column contains single imputations from the MWIAE using *self-normalized importance sampling*. Following columns contain multiple from the MIWAE using *sampling-importance-resampling*.

Table 7: Low capacity (4-layer) discriminative network for SVHN classification.

| Action (resulting layer size) |
| --- |
| Input $x$ ($32 \times 32 \times 3$) |
| Conv2D($16 \times 16 \times 16$) |
| Conv2D($8 \times 8 \times 32$) |
| Conv2D($4 \times 4 \times 64$) |
| Reshape(1024) |
| Class probabilities: Dense(10) |

For the generative models, 2 layered neural networks with 128 hidden units are used for the encoder and the decoder and a latent space of dimension 10. A Gaussian observation model is used and $K = 25$ importance samples are used during training.

We compare different imputation techniques, modelling approaches and a non-deep competitor. Zero-imputation, MissForest (Stekhoven & Bühlmann, 2012) and MICE (Buuren & Groothuis-Oudshoorn, 2010) are different imputation techniques applied, before training the discriminative model. Learnable imputation is using learnable constants as imputation, while permutation invariance is utilizing a learnable embedding. Finally, histogram-based gradient boosting (Friedman, 2001), as implemented in sklearn (Pedregosa et al., 2011), is a non-deep baseline, using the validation set for early stopping. Results are seen in figure 20.

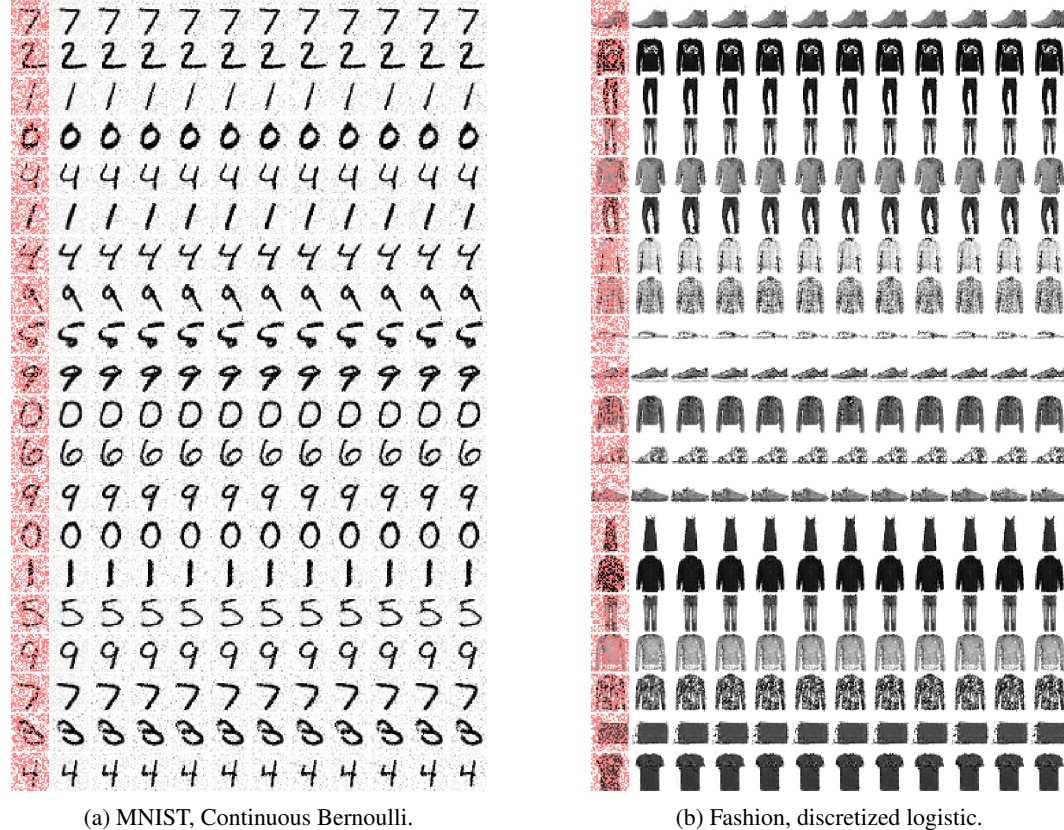

(a) MNIST, Continuous Bernoulli.          (b) Fashion, discretized logistic.

Figure 17: Missing squares. Left column contains data with missing values represented as red pixels. Second column contains single imputations from the MWIAE using *self-normalized importance sampling*. Following columns contain multiple from the MIWAE using *sampling-importance-resampling*.

Table 8: High capacity (8-layer) discriminative network for SVHN classification.

| Action (resulting layer size) |
|---|
| Input $x$ ($32 \times 32 \times 3$) |
| Conv2D($16 \times 16 \times 16$) |
| Conv2D($8 \times 8 \times 32$) |
| Gated block (Dauphin et al., 2017) ($8 \times 8 \times 32$) |
| Gated block (Dauphin et al., 2017) ($8 \times 8 \times 32$) |
| Gated block (Dauphin et al., 2017) ($8 \times 8 \times 32$) |
| Gated block (Dauphin et al., 2017) ($8 \times 8 \times 32$) |
| Conv2D($4 \times 4 \times 64$) |
| Reshape(1024) |
| Class probabilities: Dense(10) |

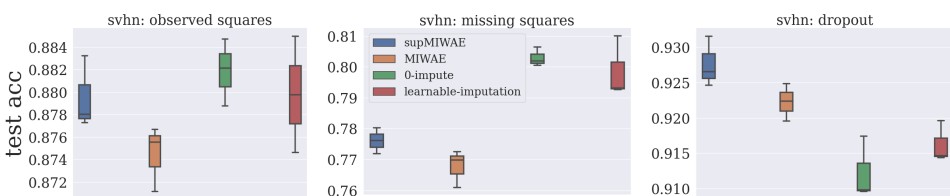

Figure 18: Test set accuracies on the SVHN dataset, with different missing mechanisms. **Left column**: observed squares. **Middle column**: missing squares. **Right column**: random dropout with missing rate 0.5.

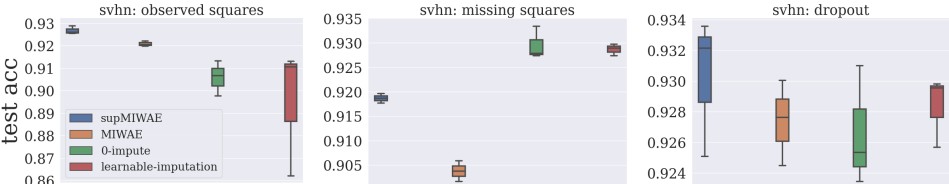

Figure 19: Fully observed test set accuracies on the SVHN dataset, with different missing mechanisms. **Left column**: observed squares. **Middle column**: missing squares. **Right column**: random dropout with missing rate 0.5.

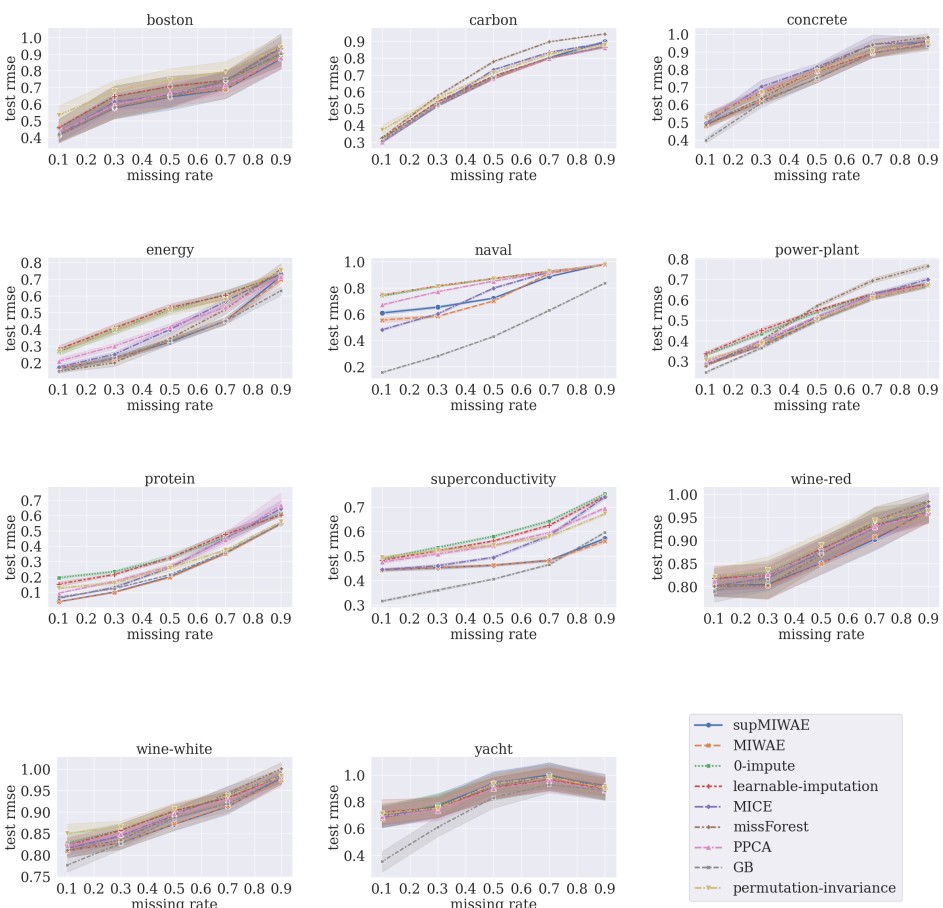

Figure 20: Test-set root mean square error on UCI datasets at varying missing rates.

