# OpenReview forum: "How to deal with missing data in supervised deep learning?"
_ICLR.cc/2022/Conference — ICLR 2022 Poster_

### Official Review · Reviewer_w981 · 2021-10-20

**Correctness:** 4
**Technical Novelty And Significance:** 3
**Empirical Novelty And Significance:** 3
**Recommendation:** 8
**Confidence:** 4

**Main Review:**

Strenghts:
- Strong theoretical foundations
- Clearly written with nice illustrations of the used principles.
- Good review of previous approaches to the supervise learning with missing data problem

Weakness:
- While the authors claim that their method is for deep learning, the provided examples are given on small datasets and used neural networks are not very deep. For example, a 4-layer CNN is used for MNIST datasets. It is not clear if the proposed method could be used on bigger datasets and deeper architectures.
- A comparison of computational time with respect to other methods is missing, which makes to think that maybe this method is too expensive to implement for big datasets and deep learning architectures.
- Experiments do not contain comparison with previous methods other than imputation based ones. There are recently published methods cited by the authors that provided implementation code for learning classifiers and missing data at the same time, i.e. avoiding the imputation followed by classification approach.


**Summary Of The Paper:**

This paper approaches the problem of supervised learning with missing data. The authors propose a probabilistic approach by jointly modeling the observed data, missing data and outcomes. The main contribution of this model is that they rely on deep generative models which seems to be an improvement from previous methods based on simpler generative models like mixture of Gaussians, for example. With the proposed model, the authors derive an optimization problem that consists in optimizing the discriminative model (classifier) and the generative model simultaneously. For the training and testing phases, the proposed algorithm considers multiple imputations of missing data, which is demonstrated to be superior to single imputation methods. Experimental results on small and simple datasets (2D datasets, MNIST digits-Fashion and regression) are nicely presented and analyzed.

**Summary Of The Review:**

A good paper including very clear and technically sound principles for solving the problem of supervised learning with missing data. Experiments are illustrative although it would be needed to implement them on bigger datasets and deeper neural network architectures as well as to compare to other methods in the literature.

---

> ### Author Response · Authors · 2021-11-22
> **Response to reviewer w981**
>
> Thank you for your review; we appreciate that you find the paper well written with nice illustrations. We will address your concerns below:
>
> > “While the authors claim that their method is for deep learning, the provided examples are given on small datasets and used neural networks are not very deep. For example, a 4-layer CNN is used for MNIST datasets. It is not clear if the proposed method could be used on bigger datasets and deeper architectures.”
>
> Thank you for this comment. To address this, we have added additional experiments on the larger  (~500.000 natural color images) SVHN dataset and with a deeper architecture. See general response.
>
> > “A comparison of computational time with respect to other methods is missing, which makes to think that maybe this method is too expensive to implement for big datasets and deep learning architectures.”
>
> The dominating computational overhead of training is the number of importance samples $K$ used in the ELBO of equation (15), and the overhead scales linearly with the number of samples. This is also what we see empirically, where one training iteration takes about 7ms for 0-imputation and learning-imputation, and about 22ms for supMIWAE with five importance samples. We have added a comment about this at the end of the introduction to section 4 (Experiment).
>
> > “Experiments do not contain comparison with previous methods other than imputation based ones. There are recently published methods cited by the authors that provided implementation code for learning classifiers and missing data at the same time, i.e. avoiding the imputation followed by classification approach.”
>
> This is a good idea, and we will add a comparison with the work by Śmieja et al (2018) in the final version of the manuscript.

---

### Official Review · Reviewer_FaAu · 2021-10-30

**Correctness:** 3
**Technical Novelty And Significance:** 2
**Empirical Novelty And Significance:** 2
**Recommendation:** 5
**Confidence:** 4

**Main Review:**

Pros:
1. The authors do a good job of explaining the challenges of training discriminative models with missing data as well as going over the related works.
2. Their approach is modular and extendable to many different choices of neural network architectures.
3. The paper is well written and easy to follow. Good work there!

Concerns and questions:
1. Are eq. 13, 14 novel contributions of this work? Or these were previously derived and new to the proposed missing value handling settings?
2. The novelty of the work is marginal. The multiple imputation idea, viewing the imputation problem as joint distributions (eq. 1,2) and also going through a few of the related works mentioned in this paper, I feel that most of the ideas proposed in this work are already out there. This work finds a combination of them which works well but the experiments are not extensive.
3. The experiments are restricted to very small datasets. Would definitely like to see how this method performs on scale. As, I can foresee some issues with the training as it uses sampling.


**Summary Of The Paper:**

The paper handles the issue of missing values in supervised deep learning settings. The fig.1 describes their method aptly. Their method (supMIWAE) is a combination of a VAE with a neural network classifier. Given the task to predict p(Y|x_{obs}, x_{miss}), the authors view it as a joint model of covariates and outcomes. Their model takes in x_{obs} and first fits a distribution to get x_{miss} and then models them as a joint distribution to predict outcome Y|(x_{obs}, x_{miss}). They use importance sampling technique to get multiple samples from the generative model.

**Summary Of The Review:**

The work is good but not novel enough. Need to show results on larger datasets. Please also highlight the fail cases in the paper.

---

> ### Author Response · Authors · 2021-11-22
> **Response to reviewer FaAu**
>
> Thank you for your review and for appreciating that the approach is modular and extendable to many different choices of neural discriminators. We address your remarks below.
>
> > “Are eq. 13, 14 novel contributions of this work? Or these were previously derived and new to the proposed missing value handling settings?”
>
> In the form, where $x^{\text{obs}}$ is the observed covariates and $y$ is the observed outcome/label, the equation is, to the best of our knowledge, new. However, we note that equations (13) and (14) are just an IWAE bound (Burda et al., 2016), where both latent and observed spaces are extended. The observed space is extended to include also the label. The latent space is extended to include $x^{\text{miss}}$ and the encoder is extended with $p(x^{\text{miss}}|z)$. Ipsen et al. (2021) have also considered an IWAE bound with an extended latent and observed space for handling MNAR values in unsupervised learning. See appendix D for more details.
>
> > “The novelty of the work is marginal. The multiple imputation idea, viewing the imputation problem as joint distributions (eq. 1,2) and also going through a few of the related works mentioned in this paper, I feel that most of the ideas proposed in this work are already out there. This work finds a combination of them which works well but the experiments are not extensive.”
>
> The idea of using multiple imputations is indeed not novel (see e.g. Rubin, 1996), and we also do not claim this. What is novel is that we jointly learn the covariate model and the outcome model, where the covariate model is a highly flexible DLVM, which, to the best of our knowledge, has not been done before. Furthermore, we also empirically illustrate the behaviour of the decision surface of the outcome model with constant imputations, which is also, to the best of our knowledge, novel.
>
> Regarding the difference with multiple imputation, we discuss it in Appendix C. Notably, a very practical difference between traditional multiple imputation and our approach, is that multiple imputation uses a uniform weight $1/K$ for each imputation, while we use importance weights that will treat each imputation based on its likelihood.
>
> In terms of extensiveness of the experiments, we would like to emphasize that we do experiments on 17 different datasets, which have various missing patterns and missing rates (UCI datasets and Image datasets), and varying model capacity (2D datasets). We have also added experiments on the SVHN dataset (see general response).
>
> > “The experiments are restricted to very small datasets. Would definitely like to see how this method performs on scale. As, I can foresee some issues with the training as it uses sampling.”
>
> Thank you for this comment. To address this, we have provided additional experiments on the larger  (~500.000 natural color images) SVHN dataset (see general response).

---

> > ### Comment · Reviewer_FaAu · 2021-11-25
> > **Follow up on the novelty of the work**
> >
> > I thank the authors for their detailed response. I followed up on the related papers pointed out and again weighed the novelty of this work. I still feel that the work is good but not novel enough and stand by my ratings. I appreciate the addition of the SVNH experiments which shows marginal improvement over other methods.

---

### Official Review · Reviewer_U6yy · 2021-10-31

**Correctness:** 3
**Technical Novelty And Significance:** 2
**Empirical Novelty And Significance:** 2
**Recommendation:** 5
**Confidence:** 3

**Main Review:**

Strengths:
+	The proposed method offers end to end training, which is useful for scalability and transferability. They claim that such a method can be applied to discriminators.
+	authors have both visually and empirically shown improvements in data imputation for MNIST and Fashion-MNIST
+	It is shown to perform better than MIWAE, (Mattei & Frellsen, 2019), the main baseline for this works

Weaknesses:
-	While there are some gains in performance, it feels incremental both in % increase and scale. MNIST and FMNIST are very similar small, almost binary valued datasets, and weakly supports the claim of scalability.
-	In UCI datasets, where features are real numbers, and probably contain more information per feature, supMIWAE performs at par with MIWAE.
-	Most aspects of the technique are leveraged from the prior works.

General comments:
●	How does the supervision and joint training impact the training complexity of the system?
●	Interestingly, supMIWAE performs definitely better than MIWAE for MNIST, and FMNIST, however such gains are no longer visible for UCI datasets. The gains of joint training seem to be limited to a certain type of data.
●	In terms of practical applications, where do the authors see their method bring the most gains? Image, language, or statistical datasets.



**Summary Of The Paper:**

The authors propose supMIWAE, a supervised missing data importance-weighted autoencoder, to address the challenges of training with missing data. They claim that the new approach, a VAE combined with a discriminator, jointly trained, is more scalable, and performs better than existing data imputation methods, in particular MIWAE (Mattei & Frellsen, 2019)

**Summary Of The Review:**

The addition of supervision and joint training to MIWAE seems novel, and while the work improves upon the prior methods of using DLVM for missing data, it falls short in the claim of scalability and overall gains in performance across different types of datasets.

---

> ### Author Response · Authors · 2021-11-22
> **Response to reviewer U6yy**
>
> Thank you for your comments and assessment of our paper. We address your individual comments below.
>
> > “While there are some gains in performance, it feels incremental both in % increase and scale. MNIST and FMNIST are very similar small, almost binary valued datasets, and weakly supports the claim of scalability.”
>
> Thank you for this question. In response, we have provided additional experiments on the SVHN dataset. See general response.
>
> > “In UCI datasets, where features are real numbers, and probably contain more information per feature, supMIWAE performs at par with MIWAE.”
> > “Interestingly, supMIWAE performs definitely better than MIWAE for MNIST, and FMNIST, however such gains are no longer visible for UCI datasets. The gains of joint training seem to be limited to a certain type of data.”
>
> The results from Le Morvan et al. (2021) and our experiments on toy datasets show that in the large capacity domain the classifier can undo any imputation and this leads to similar performance across imputation methods. The similar performance across methods on the UCI datasets is probably more related to this phenomenon than to the type of data, i.e. the characteristics of the data together with the characteristics of the classifier determines whether the supMIWAE can improve performance or not.
>
> > “Most aspects of the technique are leveraged from the prior works.”
>
> While marginalizing over missing values in a joint model of covariates and labels has indeed been approximated before (e.g. by Gharamani & Jordan, 1995), we show how to do this with highly flexible density models instead of mixtures of Gaussians. Our proposed framework could in principle also be applied to a mixture of Gaussian, so in that sense our work could be viewed as a generalization of previous methods. Similarly, our proposed importance sampling technique is general in the sense that any density model which can provide an approximation to $p(x^\text{miss} | x^{\text{obs}})$ can be plugged in as the covariate model combined with any neural classifier. Thus this approach is a generalization of previous methods (see citations at the end of section 1.1), which rely on specific density models, radial basis functions or average neuron activations for specific nonlinearities.
>
> > “How does the supervision and joint training impact the training complexity of the system? ”
>
> The training complexity of the supMIWAE is, of course, higher then just training the discriminator without missing data or with 0-imputation. In terms of training time, the dominating factor is the number of samples $K$ used in the ELBO of equation (15) and the overhead scales linearly with $K$ as compared to 0-imputation.
>
> > “In terms of practical applications, where do the authors see their method bring the most gains? Image, language, or statistical datasets.”
>
> Thank you for this interesting question. From our experiments it seems that in domains where low capacity models and inductive biases are needed for performance, the same inductive biases that are needed to improve performance in the no-missing case reduce the possibility of undoing simple imputations in the missing case, thus leaving a need for better imputation methods, such as the supMIWAE. We touch upon this in the conclusion, but we will clarify it further.

---

### Official Review · Reviewer_zqzK · 2021-11-05

**Correctness:** 4
**Technical Novelty And Significance:** 3
**Empirical Novelty And Significance:** 3
**Recommendation:** 8
**Confidence:** 3

**Main Review:**

The paper is written clearly and there are no obvious errors I can see in the theory or the experiments presented in this paper. However, I have not checked some of the derivations in detail. The method presented in the paper is largely based on previous work on variational inference, but it is otherwise novel and noteworthy theoretical contribution. Their application to train a joint model of covariates and outcomes with DLVMs and neural classifier in the presence of missing data is a fairly novel approach to my knowledge. The claims made in the paper are well backed by experiments and therefore the results are quite solid. In terms of impact, the problem of missing values is a very ubiquitous one and therefore the potential impact of the paper is significant. Moreover, the results presented by the authors indicate that their method outperforms other relevant approaches (single imputations methods) in a restricted number of scenarios: low-capacity regimes, or when the discriminative model has a strong inductive bias.

**Summary Of The Paper:**

The paper presents a method to more optimally learn deep learning classifiers in the presence of missing values. More specifically, the authors present the approach called supervised missing data importance-weighted autoencoder (supMIWAE) bound which allows them to train a joint model of covariates and outcomes by marginalizing over the missing values.

**Summary Of The Review:**

Noteworthy application of variational auto-encoders to improve classification in the presence of missing data. Overall a solid paper, I recommend accepting the paper in the conference.

---

> ### Author Response · Authors · 2021-11-22
> **Response to reviewer zqzK**
>
> Thank you for your review and feedback. We are very pleased that you find that the paper is clearly written and that the potential impact of the paper is significant!

---

### Author Response · Authors · 2021-11-22
**General response**

We would like to thank the reviewers for their valuable feedback and very useful suggestions. We appreciate the reviewers assessments that “the results are quite solid”, “the potential impact of the paper is significant”, “the paper is well written and easy to follow”, that we provided a “[g]ood review of previous approaches to the supervised learning with missing data problem” and that our technique has “[s]trong theoretical foundations”.

A common remark by the reviewers regards the scalability of the method to larger datasets and results on these with deeper architectures. We want to emphasise that we do experiments on 17 different datasets, which have various missing patterns and missing rates for the UCI datasets and Image datasets, and varying model capacity for the 2D datasets. We do these experiments to evaluate our proposed model. However, the experiments also bring new insights about the character of the decision surface. For instance, we empirically reveal the imputation manifold that was theoretically designed by Le Morvan et al. (2021). We also provide new insight on the low-capacity behaviour of the outcome model with constant imputations.

To address the reviewers’ comments about deeper architectures and larger datasets, we evaluate our method on the SVHN natural image dataset (~500.000 color images), with a low capacity (4-layers) and high capacity (8-layers) CNN discriminator. The missing mechanism “observed squares” is applied to SVHN as described in section F.2. The results are shown in table 6 of the revised manuscript and more details are given in the added section F.2. The results are also shown in the table below:

| Model   | 4-layers (test acc) | 8-layers (test acc) | 8-layers (test acc, fully obs)|
|---------|---------------------|---------------------|-------------------------------|
|supMIWAE | 0.8264              | 0.8829              | 0.9281                        |
|MIWAE    | 0.8233              | 0.8666              | 0.9143                        |
|0-impute | 0.8155              | 0.8818              | 0.8967                        |
|learnable| 0.8139              | 0.8839              | 0.9012                        |

In the low capacity regime, the supMIWAE outperforms the two single imputation methods and performs on par with MIWAE. As expected and in line with the other results in our paper, when we move to the high capacity discriminator with 8-layers, supMIWAE performs on par with the single imputation methods and outperforms the MIWAE. We observe that despite CNN’s built-in inductive bias, it can still undo single imputations when the network has a high enough capacity, which is not surprising. The last column of the table shows the test accuracy on the fully observed test set for the high capacity models trained on the training set with only observed squares. Here the supMIWAE outperforms the other models, despite it was never trained on fully observed data.

For the camera-ready version, we will for both the low- and high-capacity discriminator complete experiments for two other missing mechanisms (“missing squares” and “dropout” over a range of missing rates) and do repeated experiments to produce a figure similar to figure 6 for SVHN.

---

### Decision · Program_Chairs · 2022-01-20

**Decision:**

Accept (Poster)

**Comment:**

This paper addresses features that are missing at random in deep supervised learning, especially regression and classification. A deep latent generative model is trained in conjunction with a discriminative model, so that the distribution of the covariates is properly modeled and allows efficient variational inference for imputation.  Superior performance is achieved in low-capacity domain or when strong inductive bias is present in the discriminative model.

The paper is well written, with solid empirical support.  The approach is also generic.  Although there is some concern on novelty, overall this paper appears a solid contribution and is a good addition to the proceedings.